# An ultra-broadband photonic-chip-based parametric amplifier

Nikolai Kuznetsov[1,5], Alberto Nardi[1,2,5], Johann Riemensberger[1,4,5], Alisa Davydova[1], Mikhail Churaev[1], Paul Seidler[2 ✉] & Tobias J. Kippenberg[1,3 ✉]

Optical amplification, crucial for modern communication, primarily relies on erbium-doped fibre amplifiers (EDFAs)[1,2]. Yet, EDFAs only cover a portion of the low-loss spectrum of optical fibres. This has motivated the development of amplifiers operating beyond the erbium gain window. Pioneering work on optical parametric amplifiers (OPAs)[3,4] using intrinsic third-order optical nonlinearity has led to demonstrations of increased channel capacity. OPAs offer high gain, can reach the 3-dB quantum limit for phase-preserving amplifiers and exhibit unidirectional operation. However, power requirements for highly nonlinear fibres[3,5–8] or bulk waveguides[9,10] have impeded their adoption. By contrast, OPAs based on integrated photonic circuits offer the advantages of substantially increased mode confinement and optical nonlinearity but have been limited in bandwidth[11,12]. We overcome this challenge by using low-loss gallium phosphide-on-silicon dioxide[13–15] photonic integrated circuits (PICs) and attain up to 35 dB of parametric gain with waveguides only a few centimetres long in a compact footprint of 0.25 square millimetres. Fibre-to-fibre net gain exceeding 10 dB across an ultra-broad bandwidth of approximately 140 nm (that is, 17 THz) is achieved, with a threefold increase in the gain window compared with C-band EDFAs. We further demonstrate a high dynamic range for input signals, spanning six orders of magnitude, while maintaining a low noise figure. We exploit these performance characteristics to amplify coherent communication signals. This marks, to our knowledge, the first ultra-broadband, high-gain, continuous-wave amplification in a photonic chip, opening up new capabilities for next-generation integrated photonics.

Optical fibres and the broadband amplification of time-continuous optical signals have provided pivotal advancements in modern science and technology, particularly in the domain of optical communications for the transmission of information over long distances. A crucial breakthrough for intercontinental optical-fibre networks was the introduction of EDFAs[1,2] that could simultaneously amplify several wavelengths of light and eliminate the need for frequent electronic signal regeneration. EDFAs hence played a decisive role in the expansion of the World Wide Web. Efforts have been made to extend the bandwidth of the optical gain by using Raman-assisted EDFAs[16] and semiconductor optical amplifiers (SOAs)[17]. In contrast to amplifiers based on rare-earth-doped fibres or the more recently developed rare-earth-doped PICs[18,19], OPAs rely on the intrinsic nonlinearities of optical materials to generate gain. Such devices, also known in the literature as travelling-wave parametric amplifiers, were originally developed in the microwave domain, in which they, for example, provide quantum-limited amplification for qubit readout[20].

OPAs have unique properties that set them apart from other means of amplification[3,4]. The shape of the amplification window is entirely determined by the optical dispersion and is limited only by the transparency of the material and its nonlinear absorption threshold. The central frequency is adjustable, offering flexibility in signal processing. Parametric amplification provides the capability of frequency conversion through the creation of a concomitant idler wave carrying the same information as the input signal but at a different frequency[21,22]. The reversal of the optical phase of the idler relative to that of the signal can be used to compensate for dispersion and mitigate nonlinear effects in transmission systems[23–25]. Kerr nonlinearity provides a nearly instantaneous response, allowing rapid operation of the OPA. In contrast to the gain from erbium, parametric gain can be adjusted in situ by varying the optical-drive power while maintaining a low noise figure, a behaviour essential for amplification of weak input signals. Moreover, OPAs operate near the quantum noise limit and offer the ability to perform noiseless phase-sensitive amplification and thus have the potential to increase the span length of long-haul fibre-optic links[26,27]. Last, OPAs are unidirectional, making them resistant to optical feedback and parasitic lasing, which can readily occur with high gain.

So far, time-continuous operation of an OPA with substantial gain has only been demonstrated in fibre-based or bulk-crystal systems. For example, OPAs can be constructed by using hundreds of metres of

[1]Institute of Physics, Swiss Federal Institute of Technology Lausanne (EPFL), Lausanne, Switzerland. [2]IBM Research Europe – Zurich, Rüschlikon, Switzerland. [3]Institute of Electrical and Micro Engineering (IEM), Swiss Federal Institute of Technology Lausanne (EPFL), Lausanne, Switzerland. [4]Present address: Department of Electronic Systems, Norwegian University of Science and Technology (NTNU), Trondheim, Norway. [5]These authors contributed equally: Nikolai Kuznetsov, Alberto Nardi, Johann Riemensberger. ✉e-mail: pfs@zurich.ibm.com; tobias.kippenberg@epfl.ch

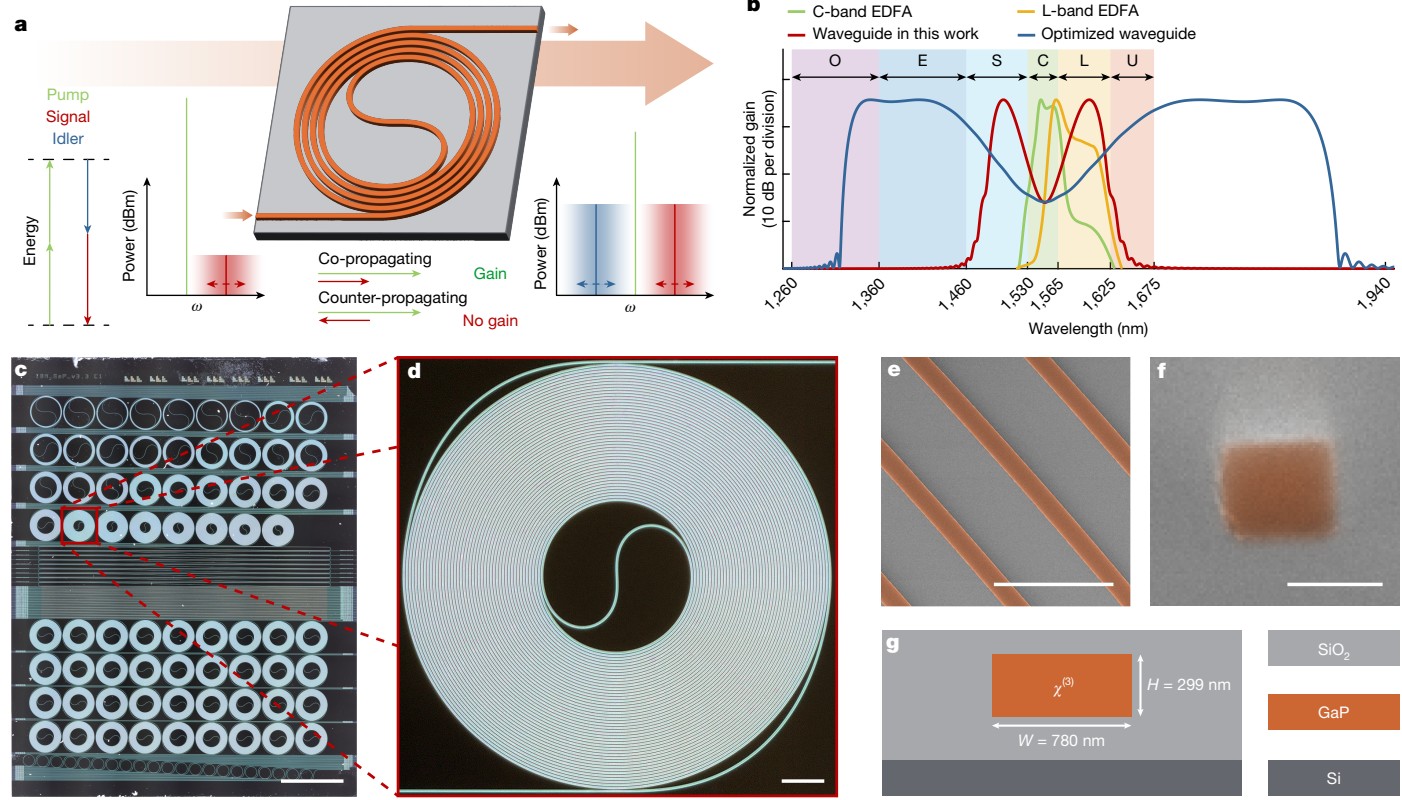

**Fig. 1 | Optical continuous-wave parametric amplification in an integrated GaP photonic waveguide. a**, Principle of broadband optical parametric amplification in an integrated spiral waveguide. Strong pump and small signal waves co-propagate through the waveguide, leading to the amplification of the signal and generation of the third wave, the idler. The frequency of the signal wave can be chosen over a wide range within the amplification bandwidth. **b**, Comparison of the typical amplification bandwidth of state-of-the-art EDFAs (green, C-band; yellow, L-band) with the amplification bandwidth of integrated GaP parametric amplifiers (red, achieved in this work; blue, if further optimized).

Gain values are provided for reference only and not meant for direct comparison. **c**, Optical microscope image of the fabricated photonic chip with several spiral waveguides and other test structures. Scale bar, 1 mm. **d**, Enlarged optical microscope image of a 5-cm-long spiral waveguide. Accounting for the chip width of 0.55 cm, the total waveguide length is 5.55 cm. Scale bar, 50 μm. **e**, Scanning electron microscope image of several waveguide turns. Scale bar, 5 μm. **f**, Scanning electron microscope image of an inverse taper facet, similar to the one used in this work, used to couple light into a GaP waveguide. Scale bar, 150 nm. **g**, Material stack of the fabricated chip.

highly nonlinear fibres[3,5–8], but they require mitigation of strong detrimental Brillouin scattering. Alternatively, mechanically cut lithium niobate waveguides[9,10] can be used, for which the $\chi^{(2)}$ nonlinearity requires periodic poling to generate quasi-phase matching and the performance is limited by the available wafer size, as the mode area is large and bending losses are excessive. By contrast, PICs and, particularly, integrated waveguides, which are equivalent to fibre OPAs, offer the possibility of a substantially increased parametric gain coefficient $g = \sqrt{(\gamma P_{\mathrm{P}})^2 - (\kappa/2)^2}$ (refs. 3,28), which is determined by the effective nonlinearity $\gamma = n_2 \omega_{\mathrm{P}}/c A_{\mathrm{eff}}$ and the phase-mismatch parameter $\kappa$; here $n_2$ is the Kerr nonlinearity, $A_{\mathrm{eff}}$ is the effective nonlinear mode area, $\omega_{\mathrm{P}}$ is the pump angular frequency and $P_{\mathrm{P}}$ is the pump power. With the appropriate choice of material, PICs can exhibit a strong Kerr nonlinearity while simultaneously featuring a high linear refractive index that substantially reduces the mode area. The material should also have negligible two-photon absorption and a high optical-damage threshold. Integrated photonic platforms that have been studied so far include silicon[29–33], chalcogenides[34], Si$_7$N$_3$ (ref. 35), AlGaAs (ref. 36) and highly doped silica[37]. None of these materials have yet produced net gain in the continuous pumping regime, as required for time-varying signals used in most applications. Recently, triggered by advances in ultra-low-loss Si$_3$N$_4$-integrated photonic waveguides[38], it was possible to achieve continuous-wave parametric amplification with net gain[11,12]. However, this approach still requires coping with high pump-power levels, metre-long waveguides, limited gain and narrow bandwidth.

Here we address these challenges by demonstrating a compact, ultra-broadband, high-gain, PIC-based OPA.

## Optical-gain measurements

We use thin-film gallium phosphide (GaP) on silicon dioxide to create an OPA comprising a dispersion-engineered waveguide operating at a pump wavelength near 1,550 nm (see Methods for details of the fabrication). GaP combines a high optical refractive index ($n = 3.05$) with a strong Kerr nonlinearity ($n_2 = 1.1 \times 10^{-17}$ m$^2$ W$^{-1}$) and an indirect bandgap[39] ($E_{\mathrm{g}} = 2.24$ eV), sufficiently large to mitigate two-photon absorption at telecommunication wavelengths. This exceptional confluence of properties has facilitated the generation of low-threshold frequency combs[14] and dissipative Kerr solitons[15]. We estimate the effective nonlinearity of the GaP waveguides in this work to be $\gamma = 165$ W$^{-1}$ m$^{-1}$, which is more than 300 times larger than the value of 0.51 W$^{-1}$ m$^{-1}$ reported for Si$_3$N$_4$ waveguides[11], facilitating a 35-fold reduction in waveguide length and a 60-fold reduction in device footprint. Advances in fabrication techniques[14,15] allow us to reduce GaP waveguide optical propagation losses to 0.8 dB cm$^{-1}$ on average within the S-band, C-band and L-band (see Methods) and achieve a high bonding yield and low defect density. These developments are pivotal for the successful fabrication of centimetre-long waveguide spirals with a footprint of only 500 × 500 μm$^2$ (Fig. 1c–g and Methods).

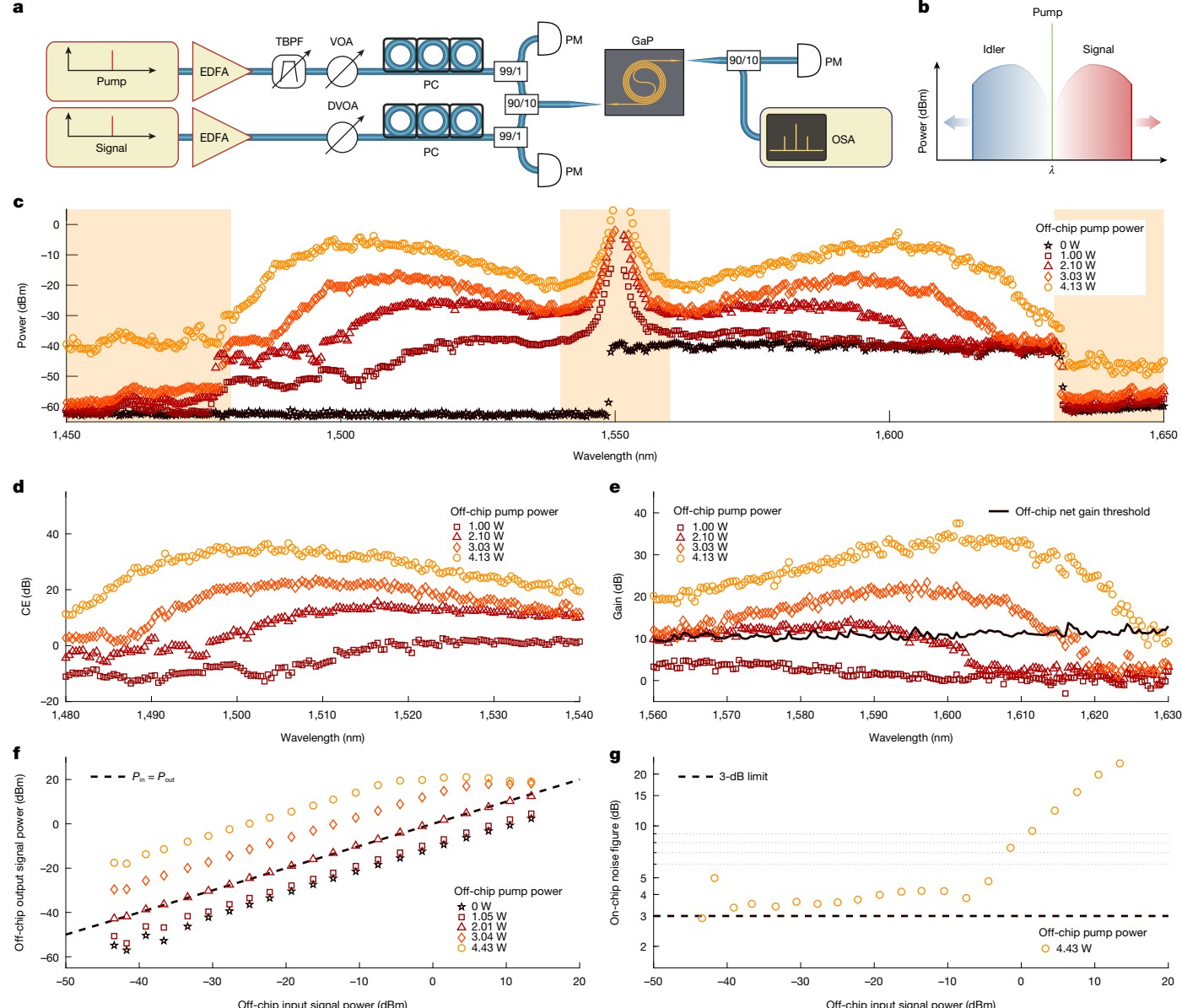

**Fig. 2 | Observation of a broadband continuous-wave parametric amplification in a GaP spiral waveguide. a**, Schematic of the experimental setup. (D)VOA, (digital) variable optical attenuator; PC, polarization controller; PM, power meter; TBPF, tunable band-pass filter. **b**, Principle of the 'Max Hold' setting of the OSA used in broadband gain measurements. With every new scan, the OSA records and updates only the highest values across the measurement span, while the wavelength of the signal laser is slowly swept to cover the whole amplification bandwidth. **c**, Measured amplification spectra for various off-chip pump powers. The resolution bandwidth of the OSA is 2 nm. EDFA and DVOA in the signal path are not used in this experiment. **d**,**e**, Calculated idler conversion efficiency (CE) and signal-amplification spectra for various input pump powers up to 4.13 W. **f**, Off-chip signal power at wavelength 1,605 nm at the output of the amplifier as a function of signal and pump powers at the input fibre. The resolution bandwidth of the OSA is 0.1 nm. The dashed line is the boundary at which the input signal power is equal to the output signal power. **g**, Measurement of the noise figure of the OPA as a function of signal input power. The dashed line indicates the 3-dB quantum limit of a phase-insensitive amplifier.

We operate the OPA with a single pump to amplify signals through degenerate four-wave mixing (FWM), exploiting the optical Kerr effect (Fig. 1a). The optical energy is redistributed by annihilating two pump photons to amplify signal photons at a separate frequency, while simultaneously generating a phase-conjugated idler at a frequency equidistant from the pump laser but offset in the opposite direction. The near-instantaneous action of the optical Kerr effect necessitates phase-matching in the waveguide for efficient amplification[3], that is, $\kappa = \Delta\beta + 2\gamma P_P$ is small, in which $\Delta\beta$ is the linear propagation mismatch given by $\Delta\beta \approx \beta_2(\omega_S - \omega_P)^2 + \beta_4(\omega_S - \omega_P)/12 +...$, $\beta_2$ and $\beta_4$ are the second-order and fourth-order derivatives with respect to the angular frequency $\omega$ of the optical propagation constant $\beta$ evaluated at the pump frequency $\omega_P$ and $\omega_S$ is the signal angular frequency. Only even-order dispersion terms contribute to the shape of the amplification spectrum. The largest optical bandwidth is achieved when the dispersion is slightly anomalous ($\beta_2 < 0$) and the linear phase mismatch $\Delta\beta$ is compensated by the nonlinear phase mismatch $2\gamma P_P$ originating from self-phase and cross-phase modulation. The peak gain in a waveguide of length $L$ is defined by $G_S = 1 + (\sinh(-\Delta\beta L_{\text{eff}}/2))^2$, in which $L_{\text{eff}} = (1 - \exp(-\alpha L))/\alpha$ is the effective length and $\alpha$ is the linear propagation loss. Figure 1b shows a comparison of the gain spectrum calculated for an optimized GaP OPA with that of C-band and L-band EDFAs. The optimized GaP waveguide has a cross-section of 789 × 299 nm², for which $\beta_2 = -16$ fs² mm⁻¹ and $\beta_4 = 3,547$ fs⁴ mm⁻¹. The region of signal

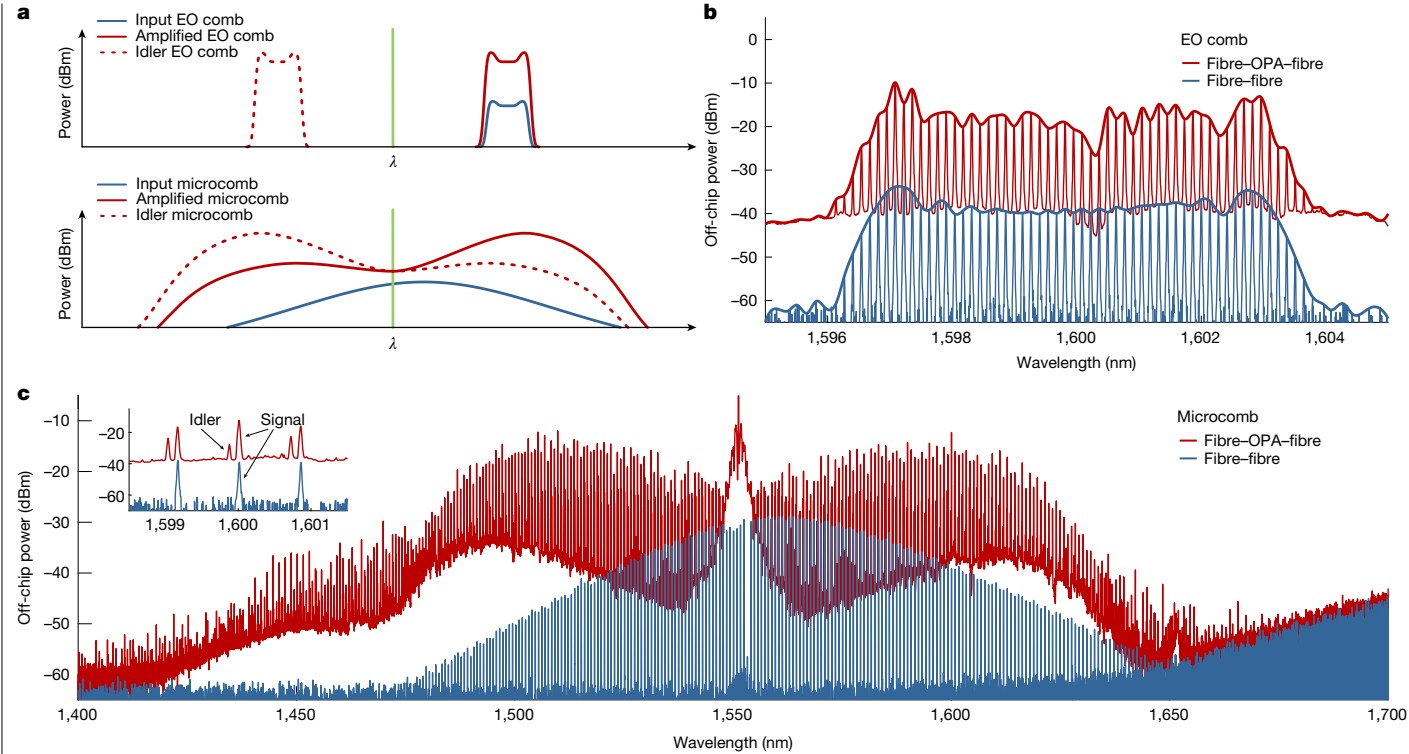

**Fig. 3 | Amplification of optical frequency combs using a GaP OPA.**
**a**, Schematic of amplification and frequency conversion of a EO frequency comb (top) and a dissipative Kerr soliton frequency comb (bottom). **b**, Spectra of the amplified (red) and input (blue) EO frequency comb showing greater than 20 dB fibre-to-fibre net gain. Envelopes are added for better visibility. **c**, Same as **b** but for amplification of a 100-GHz frequency comb produced by a single soliton state in a $Si_3N_4$ microring cavity. The inset shows the appearance of frequency-converted idler-comb lines as a result of degenerate FWM. The increased noise floor at longer wavelengths in both spectra is because of the proximity to the end of the OSA measurement range. The resolution bandwidth of the OSA is 0.02 nm in both experiments.

and idler gain extends from around 1,300 nm in the optical O-band to 1,900 nm, well beyond the longest wavelengths used in silica fibre communications. We find that the waveguides used in this work exhibit a stronger anomalous dispersion, $\beta_2 = -124$ fs$^2$ mm$^{-1}$, than expected from our simulations (see Methods). We measure the continuous-wave amplification spectrum of a 5.55-cm-long GaP spiral waveguide with a height of 299 nm and a design width of 780 nm using the same methods as described in ref. 11. Figure 2a,b depicts the experimental setup and recording scheme used to measure the parametric gain (see Methods) and Fig. 2c shows the power-calibrated amplification spectrum. In the spectral region around the pump wavelength, we observe a broadened peak resulting from the leakage of the amplified spontaneous emission (ASE) from the EDFA installed in the optical path of the pump; our tunable filter has a bandwidth of 1 nm and a finite suppression ratio of 30 dB. The frequency conversion and amplification spectra (Fig. 2d,e) are determined by comparing the output spectra recorded as the signal laser (set to 1.5 µW) is swept at various pump-power levels (red to orange) with the output spectrum obtained when the pump is switched off (black). Increasing the input power from 1.00 W to 4.13 W, we observe increasing gain, conversion efficiency and amplification bandwidth, following established theory[3,28]. As expected, we observe exponential increase in gain with a linear change in pump power. The generated idler spectrum spans from 1,550 nm to approximately 1,480 nm at the highest pump power. In Fig. 2c, spurious light generated outside the scan range of the signal laser (outer shaded regions) originates from non-degenerate FWM. Hence we only calculate the internal idler conversion efficiency (Fig. 2d) from 1,480 nm to 1,540 nm and the internal signal gain from 1,560 nm to 1,630 nm (Fig. 2e). We separately measure the passive transmission loss of the OPA, that is, the insertion loss, by bypassing the lensed fibres and the chip; the ratio

of the spectrum obtained with the bypass to the spectrum transmitted through the chip is plotted as a black line in Fig. 2e. To calculate the off-chip net gain, we subtract this 10–12 dB of insertion loss, revealing net gain of up to $G_S = 25$ dB, with $G_S > 10$ dB over a bandwidth of 70 nm. Assuming a symmetric gain spectrum on the short-wavelength side of the pump, we estimate the 10-dB off-chip gain region to exceed 140 nm, or 17 THz. This bandwidth is almost ten times larger than reported in previous work on $Si_3N_4$ (ref. 11) and nearly three times larger than the bandwidth of a single C-band EDFA. To investigate the maximum signal output power and the saturation power of the amplifier, we perform a power sweep of the signal at a wavelength of 1,605 nm (Fig. 2f). Here we use a different spiral with a designed waveguide width of 790 nm; this spiral is also used in the remaining experiments. The measurements are carried out with the resolution bandwidth of the optical spectrum analyser (OSA) set at 0.1 nm. At a pump power of 4.43 W, the saturated off-chip signal output power exceeds 125 mW, corresponding to a power of 220 mW on-chip at the waveguide output. Hence the on-chip power conversion efficiency reaches 9%, which is limited by the waveguide propagation losses. Here we also demonstrate the ability to achieve low noise amplification with high gain over a large dynamic range. The linear, small-signal regime of amplification extends up to a signal input power of almost 0 dBm at the highest pump-power level of 4.43 W. At 2.01 W pump power, the amplification is linear over the entire range of input powers measured, extending more than six orders of magnitude and limited only by the capabilities of our measurement setup; in principle, parametric amplification poses no lower limit on the signal input power. We calculate the amplifier noise figure from the measured signal amplification and the spontaneous parametric fluorescence background[40] (see Methods). The off-chip noise figure approaches 6 dB in the small-signal gain regime. Because degradation

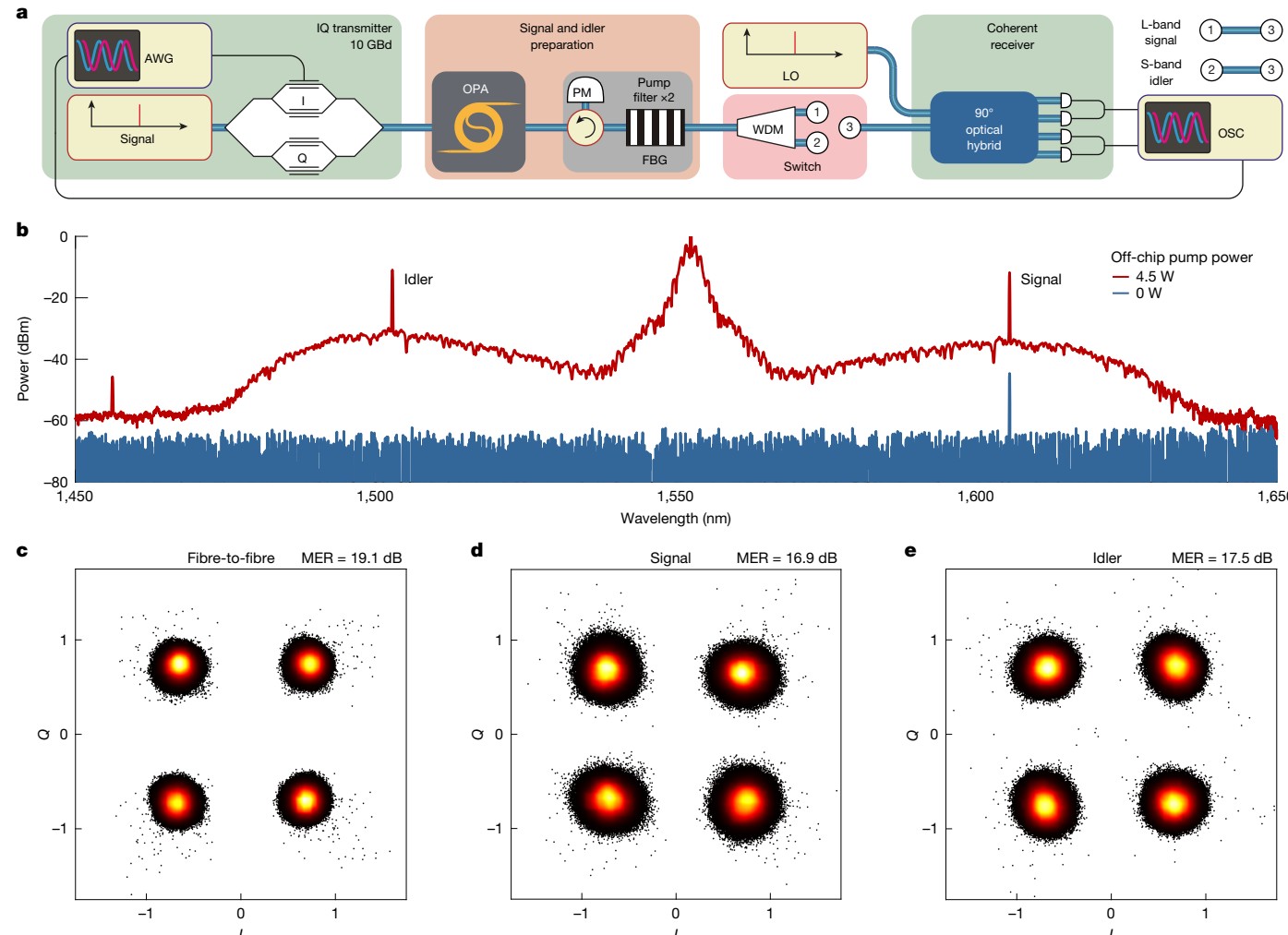

**Fig. 4 | Coherent data transmission experiment. a**, Simplified schematic of the experimental setup used in the communication experiment. AWG, arbitrary waveform generator; FBG, fibre Bragg grating; OSC, oscilloscope. **b**, Calibrated optical spectra measured at the output of the chip using the same OSA as described above (not shown here in **a**). The resolution bandwidth is set to 0.2 nm for faster acquisition. **c**–**e**, Constellation diagram and MER for three measurements: fibre-to-fibre reference, amplified signal and generated idler.

of the signal-to-noise ratio (SNR) at the input facet is irreversible, we take into account the coupling losses of 2.5 dB at the chip input facet and find an on-chip noise figure of less than 4 dB for a wide range of signal powers below saturation (Fig. 2g).

## Amplification of optical frequency combs

To highlight the application potential of such a broadband OPA, we perform further amplification experiments using low-power and high-repetition-rate frequency-comb sources as signals. In Fig. 3a, we schematically show amplification and idler generation for two types of injected frequency comb, a narrowband electro-optic (EO) comb[15] and a broadband dissipative Kerr soliton comb[41]. The EO comb is centred around 1,600 nm and has a 16-GHz line spacing and a total signal power of 10.5 µW. When transmitted through the amplifier together with a 4.12-W pump at 1,550 nm, we observe more than 20 dB fibre-to-fibre net gain (Fig. 3b). The reference dataset (blue) is measured by bypassing the lensed fibres and the photonic chip. We perform the same experiment with the frequency comb formed by a dissipative Kerr soliton state[41] generated in a $Si_3N_4$ microring resonator. The total power of the soliton comb is 76 µW and the repetition rate is 100 GHz. In the spectral region of the highest gain, far from the soliton pump located at 1,552.8 nm, the input power of individual lines decreases

to the level of several nanowatts, whereas lines with the highest initial power, located closer to the pump, experience smaller gain. As expected, an idler comb mirrored with respect to the pump frequency is generated and the line density of the output spectrum is doubled (Fig. 3c, red). A small feature in the amplified spectra around 1,651 nm is observed with all available waveguides and can be attributed to Raman scattering in GaP (refs. 14,42). The flattened spectra resulting from amplification provide clear evidence that the GaP OPA can handle the simultaneous input of several hundred lines over a broad bandwidth and the parametric gain remains as high as for a single input frequency if operated in the small-signal regime. Moreover, we again observe amplification of very small input signals, with the lowest comb lines being as weak as 1 nW, something that is not possible with typical EDFAs.

## Coherent communication and modulation transfer

To further demonstrate the potential of the GaP OPA for use in coherent communication, we set up a 10-GBd communication line, as shown in Fig. 4a. We send a quadrature phase-shift keying (QPSK)-encoded pseudo-random bit sequence at a signal wavelength of 1,605 nm (L-band) to our OPA and then perform heterodyne measurements of the amplified signal. Furthermore, using another laser as a local

oscillator (LO), we measure modulation transfer to the idler located at approximately 1,502 nm (S-band). The signal wavelength is chosen to be in the region of maximum gain, while also considering the limitations of the available equipment. Figure 4b shows the corrected OPA output spectra with the pump switched on and off. The internal gain achieved exceeds 30 dB, which is enough to compensate for coupling and propagation losses on the chip (which barely exceed 10 dB in total at the selected signal wavelength) and even for 10 dB of losses on the splitter before the chip and 4 dB of losses in the filtering section and wavelength-division multiplexer (WDM), so we achieve a positive net gain accounting for all losses present in the system (see Methods). The received signal power is 26 μW and the idler power is 32 μW; the idler power is higher than the signal power owing to lower losses at the idler wavelength. The LO power is 29.1 mW for the signal measurement and 27.7 mW for the idler. As a reference, we measure the signal directly sent from the transmitter to the receiver through a fibre, bypassing all of the other components without attenuation. We digitally post-process and analyse the data and plot constellation diagrams for all three measurements—for the fibre reference, signal and idler (Fig. 4c–e, respectively). As a figure of merit, for each measurement, we estimate the modulation error ratio (MER)[43], defined as the sum of the squares of the magnitude of the ideal symbol vectors divided by the sum of the squares of the magnitude of the symbol error vectors (see Methods). The idler conversion efficiency is high enough in the GaP OPA to generate an idler of the same power level as that of the amplified signal. This eliminates any need for further idler post-amplification, as it is sufficiently strong to be used directly for interband signal translation.

## Discussion

In summary, using an integrated GaP photonic platform, we achieve both high net gain and broadband phase-insensitive operation of an OPA. The maximum fibre-to-fibre net gain reaches 25 dB and the combined signal and idler 10-dB-gain bandwidth is 140 nm, substantially greater than the bandwidth of both EDFAs and existing continuous-wave parametric amplification systems[8,10,11]. We demonstrate the potential for practical application of GaP waveguides for the amplification of frequency combs, coherent optical data streams and optical signals over more than 60 dB of dynamic range, fully covering and extending beyond input powers from −26 dBm to 0 dBm and output powers from −8 dBm to 6 dBm, which are typical requirements in telecommunication technology[44]. The ability to amplify weak optical signals over a large dynamic range may be decisive for a variety of applications beyond coherent communications, including LiDAR[45], free-space communications and sensing. Moreover, integrated OPAs can be used where amplifiers are needed in custom bands, such as 1,650 nm for methane detection or in applications such as optical coherence tomography requiring amplification at 1,300 nm, for which no chip-scale amplifiers based on rare-earth ions exist at present. OPAs exhibit unidirectional gain, reducing the need for optical isolators in the optical path after the parametric amplifier, particularly in systems reliant on cascaded amplification chains. Although it is at present not feasible to remove the isolator from the EDFAs typically used to achieve high pump powers in OPA systems, they could potentially be replaced by semiconductor laser diodes in the future, as further reductions in optical propagation losses would greatly lower the required pump power. Our on-chip GaP OPA solves some of the most pivotal challenges that have prevented the widespread use of fibre OPA systems. Viewed more broadly, the results mark an example in which the performance of nonlinear PIC-based amplifiers surpasses that of legacy fibre-based systems. The dispersion of the waveguide can be designed as needed because it is defined lithographically, and the short length of the waveguide ensures a broad bandwidth[46] and reduces sensitivity to fabrication imperfections, in contrast to optical fibres that are hundreds of metres long. The low

Brillouin gain of integrated waveguides eliminates the technical complexity of adding pump phase modulators[7,8,26,47]. Last, the OPA can, in principle, achieve a lower noise figure than an EDFA and lower than what we have demonstrated here, as it can be operated as a phase-sensitive amplifier[12,27], reducing the noise below the quantum limit and simultaneously increasing the peak gain by 6 dB.

Our results signal the emergence of compact, high-performance PIC-based optical integrated OPAs with large bandwidth and high gain that have the potential to transition from the laboratory into future optical communication systems. Broadband OPAs may find use in a variety of future system configurations. For example, nonlinear distortions in parametrically amplified fibre links may be mitigated by alternately propagating the signal and phase-conjugated idler through successive fibre link sections[48].

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

# Methods

## Device fabrication

We fabricate the GaP photonic chip using a process similar to that outlined in the Supplementary Information of ref. 14. We epitaxially grow thin films by metal–organic chemical vapour deposition on a sacrificial [100]-oriented GaP wafer that is repolished before growth to mitigate the formation of hillocks. Hillocks form as a result of surface contamination of the as-received commercial GaP wafers and lead to bonding defects. The grown layers consist of a 100-nm-thick GaP buffer layer, a 200-nm-thick $Al_{0.2}Ga_{0.8}P$ layer (later used as an etch-stop layer to selectively remove the sacrificial wafer) and a 299-nm-thick GaP device layer. The surfaces of both the GaP wafer following growth and of a silicon wafer capped with 2 μm of $SiO_2$ are prepared for wafer bonding by depositing 5 nm of $Al_2O_3$ by thermal atomic layer deposition. After bonding, the wafer is annealed to improve the strength and stability of the bond. The sacrificial GaP wafer is then removed by mechanical grinding, down to a thickness of about 50 μm. The remaining portion of the sacrificial GaP wafer is dry-etched on the chip level in a mixture of $SF_6$ and $SiCl_4$ in an inductively coupled plasma reactive-ion etching process[49]. The etch rate slows substantially once the $Al_{0.2}Ga_{0.8}P$ layer is exposed to the plasma. Subsequently, the $Al_{0.2}Ga_{0.8}P$ stop layer is selectively removed by wet-etching in concentrated HCl for 4 mins. The surface of the chip is promptly covered with 3 nm of $SiO_2$ deposited by plasma atomic layer deposition at a temperature of 300 °C. This thin layer of $SiO_2$ also acts as an adhesion promoter for the negative resist hydrogen silsesquioxane that is used to pattern the devices by means of electron-beam lithography. The spirals are designed to fit in a single 525 × 525-μm electron-beam write field to mitigate possible scattering losses that may originate from the imperfect stitching of neighbouring write fields. The resist pattern is transferred into the GaP by inductively coupled plasma reactive-ion etching using a mixture of $BCl_3$, $Cl_2$, $CH_4$ and $H_2$ (ref. 13), after which the hydrogen silsesquioxane is removed by dipping the chip in buffered hydrogen fluoride for 10 s. A 2-μm-thick $SiO_2$ top cladding is applied by means of plasma-enhanced chemical vapour deposition using tetraethyl orthosilicate as precursor at 400 °C. Efficient input and output coupling of the light is enabled by 250-μm-long inverse tapers with a design tip width of 180 nm. The edges of the chip are removed by subsurface-absorption laser dicing to expose the inverse tapers, providing clean vertical chip facets and efficient fibre-to-chip coupling to reduce the overall optical insertion loss.

## Dispersion engineering of GaP waveguides for optical parametric amplification

In strongly confining integrated waveguides, such as those made of GaP, it is possible to tune the group velocity dispersion $\beta_2$ and the zero-dispersion wavelengths over a wide range by varying the cross-sectional geometry[50,51]. We therefore perform dispersion simulations of straight rectangular GaP waveguides fully cladded with $SiO_2$ for a range of film thicknesses and waveguide heights and widths using a commercially available finite-element-method solver, COMSOL Multiphysics, as depicted in Extended Data Fig. 1. We find that the thickness of the GaP cannot be less than about 270 nm, otherwise dispersion is normal and only a relatively weak parametric gain is possible in a narrow window near the pump wavelength.

For the prepared GaP layer thickness of 299 nm, we design a waveguide with a width of 790 nm to operate in the anomalous dispersion regime. We also fabricate waveguides with smaller widths, varied in steps of 5 nm. Note that, because of the strong mode confinement, the difference between the dispersion of a straight waveguide and that of a bent waveguide is negligible for bending radii ≳50 μm. In our Archimedean spiral waveguides, the bending radius changes from 250 μm to approximately 80 μm. It may be possible to increase the amplification bandwidth further by increasing the GaP waveguide height to about 330 nm because the contribution of the fourth-order

dispersion parameter is near zero and varies less with the waveguide width at this thickness, thus leading to more broadband amplification and a more robust waveguide design.

In our main experiments, we use waveguides that are 5.55 cm long, including the straight waveguide sections and the input and output inverse coupling tapers, as well as the spiral itself. The parametric amplification bandwidth depends on the accumulated phase mismatch between pump, signal and idler and, therefore, on the propagation constant mismatch and the device length[46]. Hence it is generally beneficial to use shorter devices to prevent bandwidth narrowing. Moreover, short devices are less sensitive to cross-section variations that could potentially lead to further bandwidth degradation.

## Numerical calculations of optical parametric gain

Numerical calculations of the signal gain $G_S$ and idler conversion efficiency $G_I$ are performed using frequency-domain nonlinear coupled-mode equations of the complex normalized amplitudes $A_{P,S,I} = \sqrt{P_{P,S,I}}$ of pump, signal and idler waves[3]

$$
\begin{aligned}
\frac{dA_P}{dz} &= (i\gamma(|A_P|^2 + 2|A_S|^2 + 2|A_I|^2) - \alpha/2) \\
&\quad \times A_P + i\gamma A_P^\star A_S A_I \times e^{i\Delta\beta z}, \\
\frac{dA_S}{dz} &= (i\gamma(2|A_P|^2 + |A_S|^2 + 2|A_I|^2) - \alpha/2) \\
&\quad \times A_S + i\gamma A_P^2 A_I^\star \times e^{-i\Delta\beta z}, \\
\frac{dA_I}{dz} &= (i\gamma(2|A_P|^2 + 2|A_S|^2 + |A_I|^2) - \alpha/2) \\
&\quad \times A_I + i\gamma A_P^2 A_S^\star \times e^{-i\Delta\beta z},
\end{aligned}
\tag{1}
$$

in which $\alpha$ denotes the linear propagation loss and $\gamma$ denotes the effective nonlinearity of the GaP strip waveguide. Here $n_2$ is the nonlinear refractive index of the waveguide core and $A_{eff}$ is the effective nonlinear mode area:

$$
\gamma = \frac{\omega_P n_2}{c A_{eff}}, \quad A_{eff} = \frac{\left(\int |\mathbf{E}|^2 \, dA\right)^2}{\int |\mathbf{E}|^4 \, dA}.
\tag{2}
$$

All simulations are implemented in MATLAB (version 9.13.0 (R2022b)), whereas dispersion, effective nonlinearity and the effective nonlinear mode area are calculated using COMSOL Multiphysics. For more precise simulations of broadband gain, the variation of mode profile with frequency should be taken into account; the effective nonlinear mode area should be replaced with signal-frequency-dependent overlap integrals of the nonlinear interaction[52]. For waveguides with a width of 790 nm and height of 299 nm, we find an effective nonlinear mode area of 0.26 μm² and an effective nonlinearity $\gamma$ of 165 W⁻¹ m⁻¹ at a pump wavelength of 1,550 nm, with the nonlinear refractive index $n_2 = 1.1 \times 10^{-17}$ m² W⁻¹. This value of $\gamma$ is more than 300 times larger than that of dispersion-engineered $Si_3N_4$ waveguides[11] and more than $10^4$ times larger than typical highly nonlinear fibres[8]. The nonlinear coupled-mode equations are integrated using a forward Euler scheme along the waveguide spiral of length $L$ and signal gain and idler conversion efficiency are calculated relative to the input signal power $P_S(0)$:

$$
G_S(L) = \frac{P_S(L)}{P_S(0)}, \quad G_I(L) = \frac{P_I(L)}{P_S(0)}
\tag{3}
$$

In the small-signal regime, the maximum gain in a waveguide of length $L$ is given by $G_S = 1 + (\gamma P_P g^{-1} \sinh(gL))^2$ and achieved when the linear propagation mismatch $\Delta\beta = \beta(\omega_S) + \beta(\omega_I) - 2\beta(\omega_P) \approx \beta_2(\omega_S - \omega_P)^2 + \beta_4/12(\omega_S - \omega_P)^4 + \dots$ is compensated by the nonlinear phase-mismatch $2\gamma P_P$ and, therefore, is defined by $\Delta\beta + 2\gamma P_P = 0$ and $g = \gamma P_P$, yielding $G_S = 1 + (\sinh(-\Delta\beta L/2))^2$. Extended Data Fig. 2a shows simulated amplification spectra for the optimized waveguide cross-section.

Our devices turned out to feature a more strongly anomalous dispersion than simulated, resulting in the gain profile depicted in Extended Data Fig. 2b. This discrepancy may originate from imperfections and variations in the waveguide geometry and material stack that are not considered in the simulations. The waveguide cross-section for the simulations in Extended Data Fig. 2b has been adjusted to match experimental results, giving a $\beta_2$ parameter of $-120$ fs$^2$ mm$^{-1}$, whereas $\beta_4$ has a negligible contribution. For the nominal cross-section of 789 × 299 nm$^2$, the theoretical value would be $\beta_2 = -16$ fs$^2$ mm$^{-1}$ (and $\beta_4 = 3{,}547$ fs$^4$ mm$^{-1}$). The experimentally measured value is $-124$ fs$^2$ mm$^{-1}$ (Extended Data Fig. 3), which is in a good agreement with numerical simulations for the adjusted amplification spectra, given that the spiral waveguide is short and precise measurements are challenging. The value of $\beta_4$ cannot be reliably determined experimentally.

We are at present performing more measurements to determine the reason for the discrepancy between the experiment and simulations and to improve the material model of GaP, which is pivotal to achieve the full potential amplification bandwidth of 500 nm. However, we are ultimately limited by fabrication tolerances. Extended Data Fig. 4 shows that the parametric gain spectra can vary substantially if the cross-section of the fabricated waveguide deviates only slightly from the design. Although variations of several nanometres in waveguide width may be acceptable, a deviation in height of only 3 nm changes the gain profile substantially.

In the literature, there is great uncertainty in the value of the nonlinear refractive index of GaP, ranging from $n_2 = 2.5 \times 10^{-18}$ W$^{-1}$ m$^2$ (ref. 53) measured by FWM to $n_2 = 1.1 \times 10^{-17}$ W$^{-1}$ m$^2$ (ref. 14) measured by means of the optical parametric oscillation threshold in GaP ring resonators, as well as by modulation transfer experiments. We find that, by using the smaller value, we underestimate the observed gain substantially. Using the larger value, we need to reduce the input power in the simulation by 3 dB. A similarly reduced Kerr parametric gain was recently observed in Si$_3$N$_4$-based waveguide spirals[11,12]. The first reason for this observation may be residual higher-order nonlinear absorption in the GaP waveguide, as we operate the amplifier at high input power and close to the three-photon absorption threshold. Moreover, Ye et al. have proposed that the reduced effective pump power arises owing to parasitic power transfer between different waveguide modes[12]. This is corroborated by the observation of substantial modulation of the transmission spectrum owing to chaotic interference of the fundamental and higher-order modes (see Extended Data Fig. 5a and the discussion below), proving that at least some of the optical power is propagating in the higher-order mode. Overall, we find that our observations support a value for the nonlinear refractive index towards the upper limit of the literature range.

## Transmission characterization and loss measurements

The dispersion profile and propagation loss of the spiral are measured with a custom, all-band frequency-comb-calibrated scanning diode laser spectrometer and optical frequency-domain reflectometer (OFDR)[54] over the wavelength range from 1,260 nm to 1,630 nm (refs. 11,55). We characterize the transmission of our samples at low power, as described in ref. 11. A fully calibrated transmission trace of a 5.55-cm-long spiral waveguide is shown in Extended Data Fig. 5a.

The transmission exhibits a strongly oscillating behaviour, which we attribute to the interference of the facet reflections and multimode interactions in the waveguide. At wavelengths around 1,550 nm, the overall transmission is 12%. However, during OPA gain measurements, the photonic chip is exposed to optical powers as high as 4.4 W and the transmission trace is thermally redshifted. To maintain a good coupling of the pump laser when increasing the power, we adjust the pump wavelength, increasing it typically by not more than 1.6 nm at the highest available level of pump power. In the S-band, C-band and L-band, the average loss rate is 0.8 dB cm$^{-1}$ (Extended Data Fig. 5b). At shorter wavelengths, we observe increased losses that we attribute to absorption by the first overtone of the O–H stretch vibration in the low-temperature oxide cladding, which can be mitigated by using different fabrication techniques[56]. From these measurements, we estimate an average coupling loss rate as high as 2.5 dB per facet, assuming that the input and output facets are the same.

## Optical-gain measurements

We use two widely tunable external-cavity diode lasers (TOPTICA CTL) as pump and signal sources. The pump laser is amplified using an EDFA (Keopsys CEFA-C) up to 4 W. The ASE from the EDFA is filtered out with a tunable band-pass filter (Agiltron FOTF) and the input power is controlled with a variable optical attenuator (Schäfter + Kirchhoff 48AT-0). One percent of each of the input waves is guided to power meters (Thorlabs S144C) and the rest is combined on a 10/90 fibre splitter, with the pump being injected in the 90% input. We use lensed fibres to couple light into and out of the waveguide. Of the collected light, 90% is guided to the power meter and 10% is analysed using the OSA (Yokogawa AQ6370D). All input (output) powers quoted in the figures and the text are calibrated and indicate the values at the input (output) lensed fibre tips, unless specified separately. The pump wavelength is set to 1,550 nm. For each pump-power level, we continuously scan the signal laser wavelength from 1,550 nm to 1,630 nm (which is the maximum available wavelength for the laser that we use) while simultaneously scanning the OSA using the 'Max Hold' function; at every new scan, the OSA records and updates only the highest values across the measurement span, while the signal laser wavelength is slowly swept to cover the entire amplification bandwidth. To ensure that measurements at different wavelengths are performed under the same conditions without marked coupling degradation during the experiment, we slow down the laser scan speed and set the OSA resolution bandwidth to 2 nm. We find that, in the optical-amplification experiments, it is paramount to ensure good thermal coupling between the photonic chip and the chip holder to avoid excessive heating of the waveguide that leads to device failure and burning of the waveguide in the inverse taper section, even at input powers as low as 1 W. We attribute this behaviour to the fact that the pump wavelength is very close to the three-photon absorption threshold ($\approx$1,660 nm for $E_g = 2.24$ eV (ref. 39)) and that, therefore, the nonlinear absorption of GaP increases strongly with temperature in the telecommunication wavelength region. Similar behaviour was observed in silicon[57] at telecommunication wavelengths and for GaP close to the two-photon absorption threshold[58]. However, by ensuring good thermal contact between the chip and metallic chip holder, we can operate the OPA at room temperature and at 4.1-W input power for hours in air without active cooling.

## Parametric gain in waveguides with varying lengths

As well as the continuous-wave amplification experiments presented in the main text, we explored the behaviour of the parametric gain and the conversion efficiency in spiral waveguides with the same design cross-section (waveguide width 780 nm) but different waveguide lengths, varying the length in steps of 1 cm from 2.55 cm to 5.55 cm. The conversion efficiency and parametric gain data are depicted in Extended Data Fig. 6a,b, respectively.

Here the pump power is set to 3 W. As expected, the amplification bandwidth decreases with increasing waveguide length owing to the accumulated linear phase mismatch, in good agreement with established theory[3] and previous observations[46]. For the shortest length, 2.55 cm, we find that the net amplification bandwidth extends substantially beyond the tuning range of our laser.

## Spontaneous parametric comb formation

In doped waveguide and fibre lasers, the achievable amplification gain is limited by spontaneous lasing through parasitic reflections from chip facets and other components. For example, in recent work on erbium-doped waveguide amplifiers[18], an off-chip net gain of 26 dB was

achieved but required the use of index-matching gel at the fibre facet to suppress parasitic lasing. By contrast, the strong yet unidirectional gain of the GaP OPA allows us to achieve the same net gain with simple cleaved chip facets and lensed fibres without parametric lasing because the threshold for the spontaneous sideband formation is intrinsically large compared with doped amplifiers with bidirectional gain. However, operating the amplifier at 4.43 W, we occasionally observe spontaneous parametric oscillations in the waveguide spiral (Extended Data Fig. 7), even in the absence of any input signal. Owing to the finite reflections from chip facets, amplified noise photons induce a coherent build-up, forming strong waves located at the maxima of the parametric gain lobes. A similar behaviour was recently observed in an optical fibre with single-sided Bragg reflectors[59] owing to Rayleigh scattering in the fibre.

Once formed, the comb can be present for several minutes and disappears only when the coupling degrades too much. A more rigorous investigation of the observed comb formation is out of the scope of this work and will be reported elsewhere. This measurement demonstrates that our amplifier operates at or close to its internal gain limit, yet the achievable gain may be substantially improved if facet reflection is suppressed with the same methods as used in ref. 18.

### Signal power sweep and noise figure measurements

To vary input signal power, we use an L-band EDFA (Keopsys CEFA-L) and a digital variable optical attenuator (DVOA; OZ Optics DA-100). These two components are installed in the signal optical path and used only for this experiment. We set the signal EDFA to the maximum achievable output power and change the attenuation of the DVOA over the entire accessible range, from 60 dB to 0 dB. For reference, we measure input signal power in the optical fibre before the OPA directly using the OSA in the absence of the pump wave (Extended Data Fig. 8a). The amplified signal at a pump power of 4.43 W is shown in Extended Data Fig. 8b. We estimate the noise figure using the following relation[12,40]:

$$\text{NF} = \frac{1}{G} + 2\frac{\rho_{\text{ASE}}}{Gh\nu}, \tag{4}$$

in which $G$ is optical gain and $\rho_{\text{ASE}}$ is the noise-power density, assuming a bandwidth of 0.1 nm. We note that we use a corrected bandwidth for the measurements of the noise-power density and we account for 2.5 dB of coupling losses at each chip facet. The optical SNR of the input signal is larger than 48 dB. Hence the contribution of the input ASE noise of the signal is negligible for small input signal powers, which can be seen in Extended Data Fig. 8b; the noise-power levels for the first input signal powers are essentially the same, making it possible to directly use the corresponding value as a true value for the noise power and eliminating the need to manually correct for the input signal ASE. The degradation of the noise figure at higher signal input powers is attributed to gain saturation; in other words, the signal gain becomes weaker while the parametric fluorescence is still within the small-signal regime and undergoes a higher amplification. The signal noise here originates mostly from the L-band EDFA and noise correction at high input signal powers is tricky because of the coupling fluctuations, spectral features in the transmission function of the device, higher-order FWM processes and gain saturation, leading to different gains experienced by the signal and its noise.

### Frequency-comb amplification—extended data

In Extended Data Fig. 9, we show the extended datasets for the EO comb and Kerr soliton comb amplification measurements. Note that, because of the nearly instantaneous nature of parametric amplification, we do not use a dispersive fibre compression stage during the EO comb preparation before the amplifier to avoid high signal peak powers and amplifier saturation. Because we tune the pump wavelength at each pump-power level to match the thermal drift of the spiral and to maintain good coupling, in the case of pumping with the Kerr soliton

comb, the idler comb has a different frequency offset every time. The idler comb can be seen in the data for the pump powers of 1.01 W and 1.99 W, whereas—for higher powers—the offset between the OPA pump and soliton pump is almost a multiple of the comb free spectral range, and idler comb lines are located close to the signal comb lines. Further lines can be observed throughout the entire spectra and, most notably, closer to the pump; we attribute their appearance to numerous FWM processes between various interacting lines. Further lines that are located at frequencies far outside the bandwidth of the input comb arise owing to cascaded non-degenerate FWM processes.

### Coherent communication

We use four different external-cavity diode lasers (TOPTICA CTL) as sources of emission in this experiment—pump, signal and two LOs near the signal and idler wavelength—to perform heterodyne measurements. The QPSK modulation format at 10 GBd symbol rate is used to encode a pseudo-random bit sequence in the signal using an IQ modulator (iXBlue MXIQER-LN-30) driven by an arbitrary waveform generator (Keysight M8195A) and a bias controller (iXBlue MBC-IQ-LAB-A1). Next, the modulated signal is transmitted to the OPA system, in which it is combined with a pump on a 10/90 fibre splitter, attenuating the signal by 10 dB. Just before the OPA, the signal power in the fibre measures 0.5 µW. We direct 10% of the OPA output to the OSA, whereas the remainder proceeds along the primary measurement path. We use a pair of filters, each containing a circulator and a fibre Bragg grating, to eliminate the residual pump in the fibre after the OPA section. These filters allow only the signal and idler to advance through the communication line, attenuated by only 4 dB as they pass through. We use a C + L edge-band WDM with a suppression ratio greater than 25 dB to split the signal and idler; we separately verified the normal operation of our WDM in the S-band. To perform measurements using a coherent receiver (Finisar CPRV2222A-LP installed in EVA-KIT CPRV2XXX), we switch between signal and idler by simply changing the LO source and plugging the corresponding fibre output of the WDM into the input of the receiver. We select the signal wavelength to be within the region of maximum gain and to be compatible with the capabilities of our equipment. The LO power levels are set to the maximum output of their respective lasers, after confirming that these values fall within the specified input range of the coherent receiver. The analogue signal from the coherent receiver is collected using a fast oscilloscope (Teledyne LeCroy SDA8Zi-A) at a sampling rate of 40 Gs s⁻¹ (that is, four samples per symbol) and the obtained data are digitally processed using MATLAB Communications Toolbox (version 7.8 (R2022b)) and custom functions. The names of the functions are specified in parentheses below. First, the imbalance in the collected IQ data are compensated using the Gram–Schmidt orthogonalization procedure[60] to ensure that the I and Q components of the signal are orthogonal, zero-mean and normalized. Next, we apply the coarse frequency offset compensation (comm.CoarseFrequencyCompensator) and use the automatic gain control (comm.AGC) to remove amplitude variations. The signal is decimated using a raised-cosine finite impulse response filter (comm.RaisedCosineReceiveFilter) with the decimation factor of 2. After this step, the timing synchronization (comm.SymbolSynchronizer) using the Gardner timing error detector is added to the data-processing algorithm to correct timing errors and the signal is decimated again, reaching one sample per symbol. We use the carrier synchronizer algorithm (comm.CarrierSynchronizer) to correct carrier frequency and phase offsets for accurate demodulation of the received signal. As a last step, we normalize the signal and equalize it using decision feedback filtering (comm.DecisionFeedbackEqualizer) with the constant modulus algorithm. To gain an understanding of the quality of the obtained constellations, we evaluate the MER[43] defined as

$$\text{MER} = 10\log_{10}\left(\frac{\sum_{k=1}^{N}(I_k^2 + Q_k^2)}{\sum_{k=1}^{N}(e_k)}\right), \tag{5}$$

in which $e_k = (I_k - \tilde{I}_k)^2 + (Q_k - \widetilde{Q}_k)^2$ is the squared amplitude of the error vector, $\tilde{I}_k$ and $\widetilde{Q}_k$ are measured in-phase and quadrature components of symbol vectors and $I_k$ and $Q_k$ are ideal reference values (comm. MER). The MER measures the spread of the symbol points in the constellation clusters. A wider spread results in a lower MER and lower signal quality. In the absence of notable signal degradation, the average measured symbol vector for each constellation point should coincide with the ideal symbol vector. In this case, MER becomes equivalent to the constellation SNR (or symbol SNR), which is calculated in a similar way but uses the average symbol vectors instead of the ideal reference points.

## Data availability

All experimental datasets and scripts used to produce the plots in this work are available in a Zenodo repository at https://doi.org/10.5281/zenodo.14180033 (ref. 61).

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

**Acknowledgements** We fabricated the sample at the Binnig and Rohrer Nanotechnology Center (BRNC) at IBM Research Europe – Zurich. This work was supported by the European Union's Horizon 2020 research and innovation programme under the Marie Skłodowska-Curie grant agreement no. 812818 (MICROCOMB), by the Swiss National Science Foundation (SNSF) under grant number 216493 (HEROIC) and by the Air Force Office of Scientific Research under award number FA9550-19-1-0250.

**Author contributions** N.K. and J.R. performed numerical simulations of dispersion and parametric gain and designed spiral waveguides. A.N. fabricated the sample. J.R. and N.K. performed the initial linear characterization of the devices. N.K. and A.N. carried out the broadband gain measurements. N.K. and A.D. conducted frequency-comb amplification experiments. N.K. and M.C. executed the communication experiment. N.K. carried out the rest of the measurements presented in this work. N.K. and J.R. prepared the manuscript, with contributions from all authors. J.R., P.S. and T.J.K. supervised the work.

**Funding** Open access funding provided by EPFL Lausanne.

**Competing interests** J.R., N.K. and T.J.K. are inventors on European patent application no. 24165690.9 submitted by the Swiss Federal Institute of Technology Lausanne (EPFL) that covers "Optical parametric amplifier apparatus and optical signal transmission apparatus, and applications thereof". The other authors declare no competing interests.

**Additional information**
**Correspondence and requests for materials** should be addressed to Paul Seidler or Tobias J. Kippenberg.

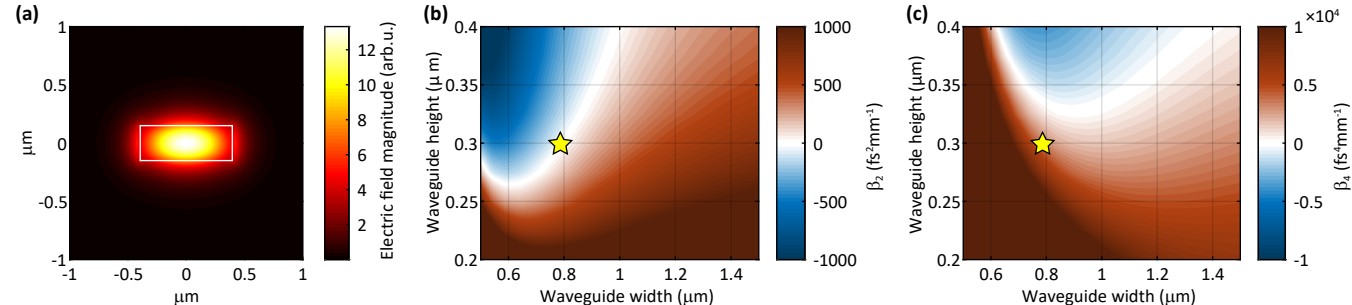

**Extended Data Fig. 1 | Dispersion of integrated GaP waveguides. a**, Electric-field profile of the fundamental TE mode in a rectangular GaP waveguide with a width of 790 nm and a height of 299 nm. The mode is strongly confined owing to the high refractive index contrast between the GaP core and the $SiO_2$ cladding. **b**,**c**, Second-order and fourth-order dispersion maps, respectively. The anomalous dispersion is available for waveguides with a height greater than 270 nm. Stars indicate the regions of optimal parameters necessary to achieve the broadband operation of the optical travelling-wave parametric amplifier made on a chip with a predefined GaP thickness of 299 nm. Although slightly anomalous dispersion is typically required for parametric amplification because it compensates for nonlinear phase mismatch, the fourth-order dispersion term should be positive to counteract the excess propagation phase mismatch that accumulates as a result of second-order dispersion, thereby achieving broadband operation.

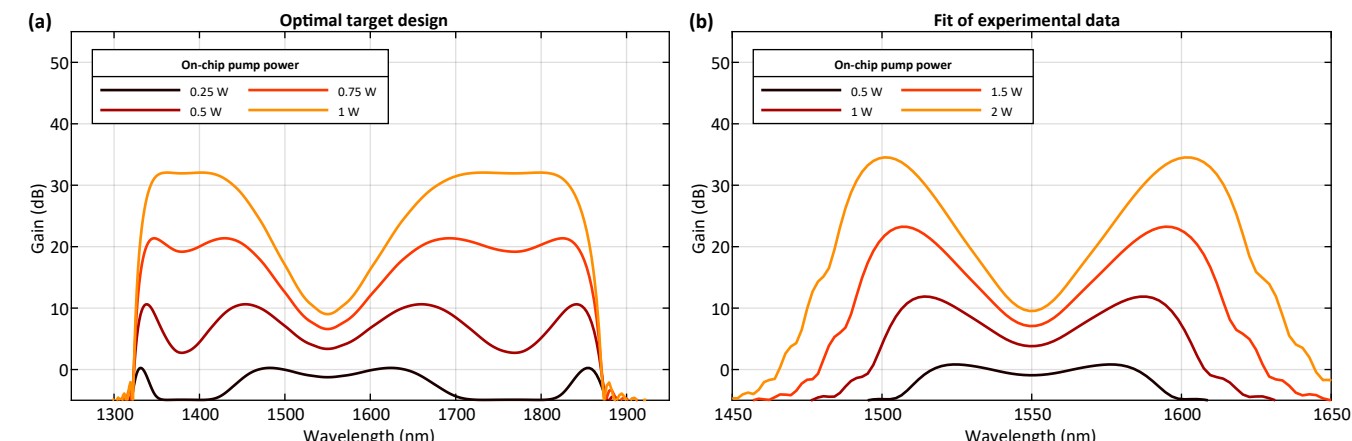

**Extended Data Fig. 2 | OPA gain simulations. a**, Gain in the waveguide with a theoretically optimal cross-section of 789 × 299 nm². At lower pump powers, two more distinct gain peaks appear separate from the main part of the gain spectrum and their positions correspond to the frequencies at which the nonlinear phase-matching condition is satisfied with the help of the fourth-order dispersion term. These peaks merge with the central part as the pump power increases. **b**, Gain in the waveguide with dispersion and nonlinearity tuned to match our experimental observations. Owing to strong anomalous dispersion, the contribution of the fourth-order dispersion term diminishes, causing the two more gain peaks to move well beyond the simulation wavelength window. Hence the gain profile contains only two main gain lobes.

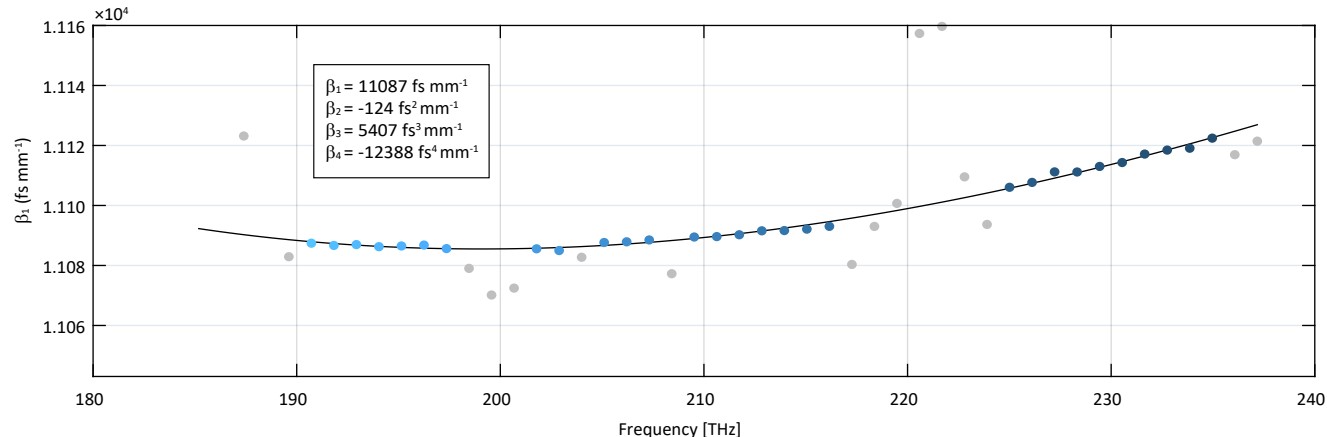

**Extended Data Fig. 3 | Dispersion measurements of a 790-nm-wide spiral waveguide using frequency-comb-assisted spectroscopy.** Dispersion measurements of a GaP waveguide are performed using OFDR by analysing the delay between reflections from the input and output facets at different frequencies to determine the dispersion characteristics of the waveguide. Grey dots are excluded from the dataset before fitting owing to their large deviation. The values for $\beta_3$ and $\beta_4$ are given for reference and are not reliable because of the inaccuracy of the dispersion measurements of an extremely short waveguide.

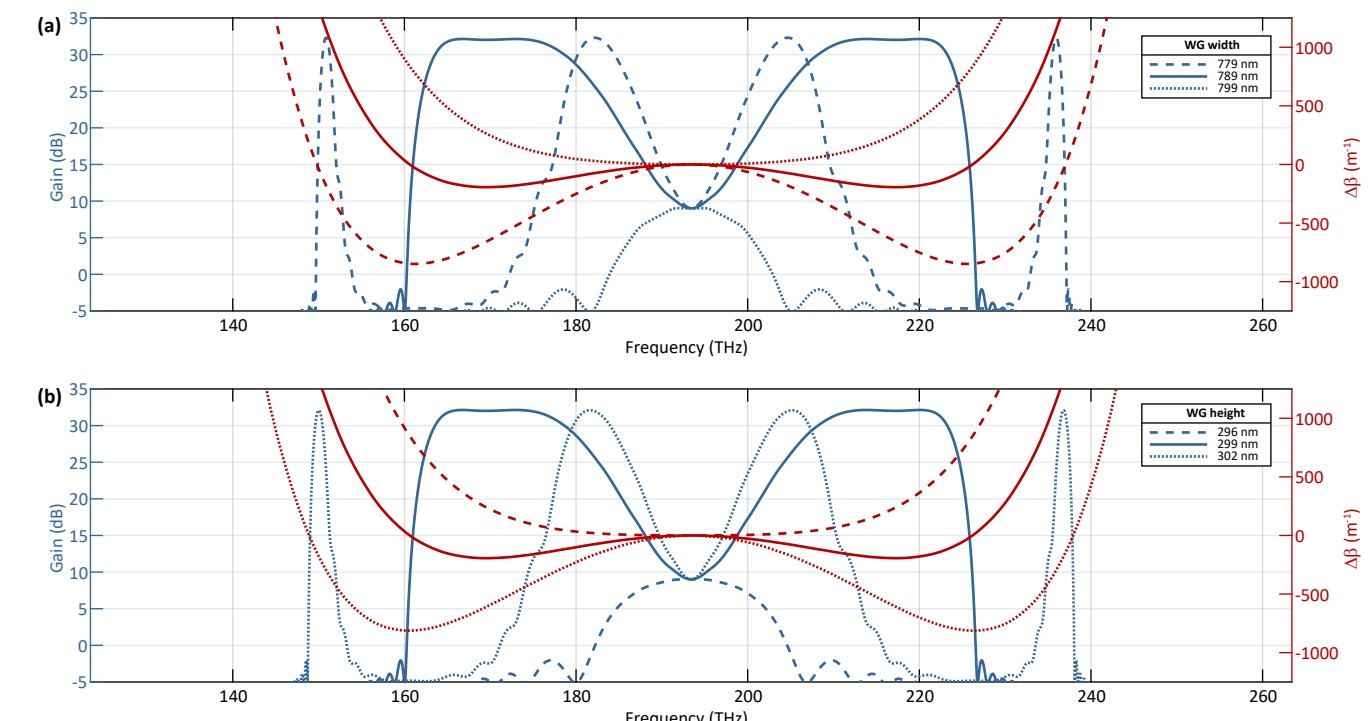

**Extended Data Fig. 4 | Variations in gain spectra and propagation mismatch owing to fabrication tolerances. a**, Waveguide height is 299 nm, with waveguide width varied by ±10 nm. A wider waveguide has normal dispersion and therefore substantially reduced gain. **b**, Waveguide width is 789 nm, with waveguide height varied by ±3 nm. A waveguide with a smaller height exhibits normal dispersion. The sensitivity of the dispersion profile is more pronounced compared with variations in the waveguide width. Variations in the cross-section that result in more anomalous dispersion do not notably affect the effective nonlinearity and the maximum gain but only lead to a reduction in bandwidth.

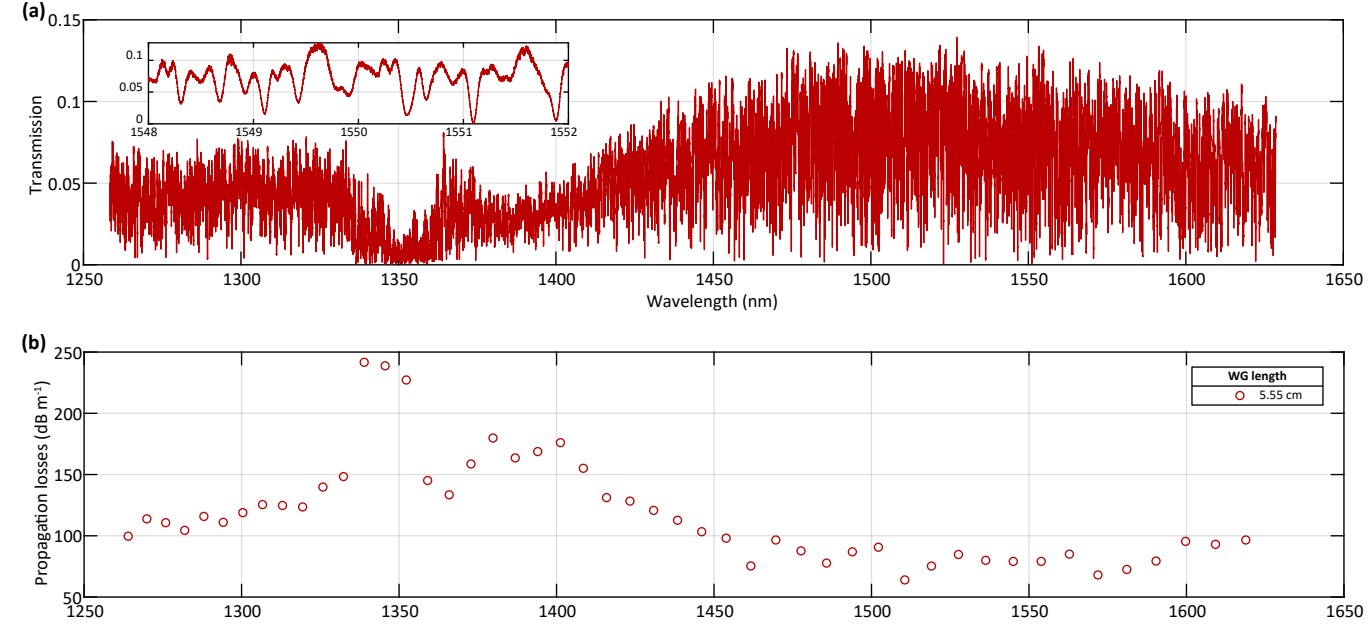

**Extended Data Fig. 5 | Loss measurements of a GaP waveguide. a**, Calibrated low-power transmission trace of the spiral waveguide with waveguide width of 790 nm. The strong oscillations are attributed to higher-order mode interference, whereas the reduced transmission around 1,350 nm is caused by hydrogen absorption. Inset, zoomed-in view of the region in which the pump wavelength is typically set in our experiments. In certain regions, the total waveguide transmission can reach up to 12%. **b**, OFDR measurements of the same device, indicating frequency-dependent propagation losses. Increased losses at 1,350 nm are consistent with previous observations of reduced transmission around this wavelength.

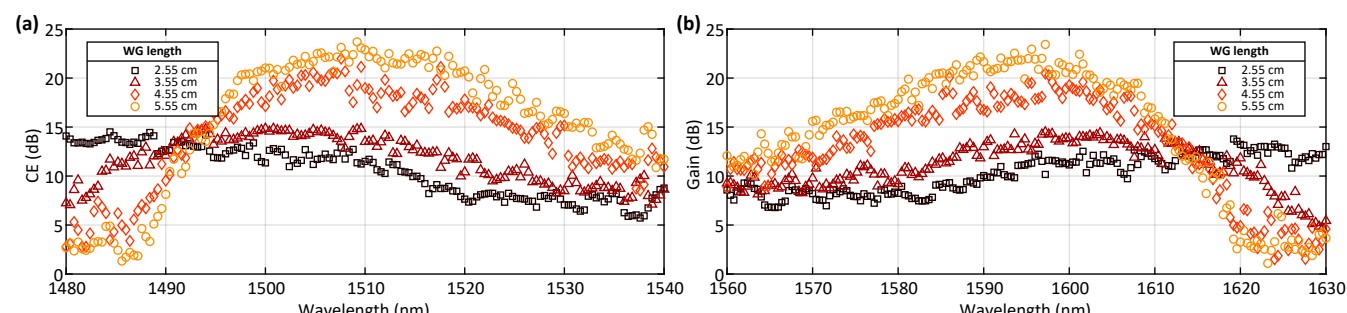

**Extended Data Fig. 6 | Parametric amplification and idler conversion efficiency as a function of waveguide length. a**, Idler conversion efficiency in waveguides with lengths ranging from 2.55 cm (black squares) to 5.55 cm (orange circles), recorded with a pump power of 3 W. All waveguides have cross-sections of 780 nm × 299 nm. **b**, Parametric amplification spectra for the same waveguides. Both conversion efficiency and parametric gain increase with waveguide length, whereas the bandwidth decreases as a result of accumulated phase mismatch.

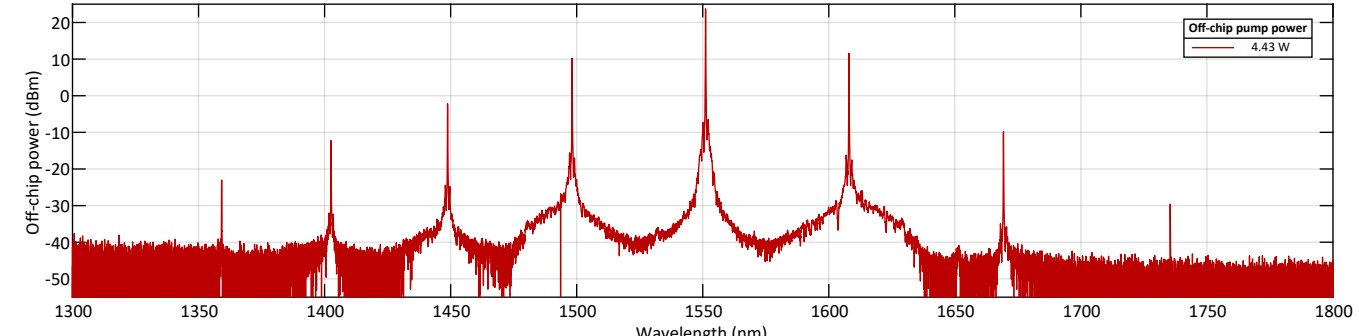

**Extended Data Fig. 7 | Parametric comb formation in the waveguide with only the pump wave injected.** The pump power is set to 4.43 W; the positions of the first sidebands correspond to the maxima of the gain lobes. The spectrum is stable as long as the pump power is maintained high enough and the fibre-to-chip coupling is optimized. A small feature at 1,650 nm indicates the Raman effect. As the data were captured using a different OSA (Yokogawa AQ6375), the noise floor seems different compared with the spectra presented elsewhere in this work.

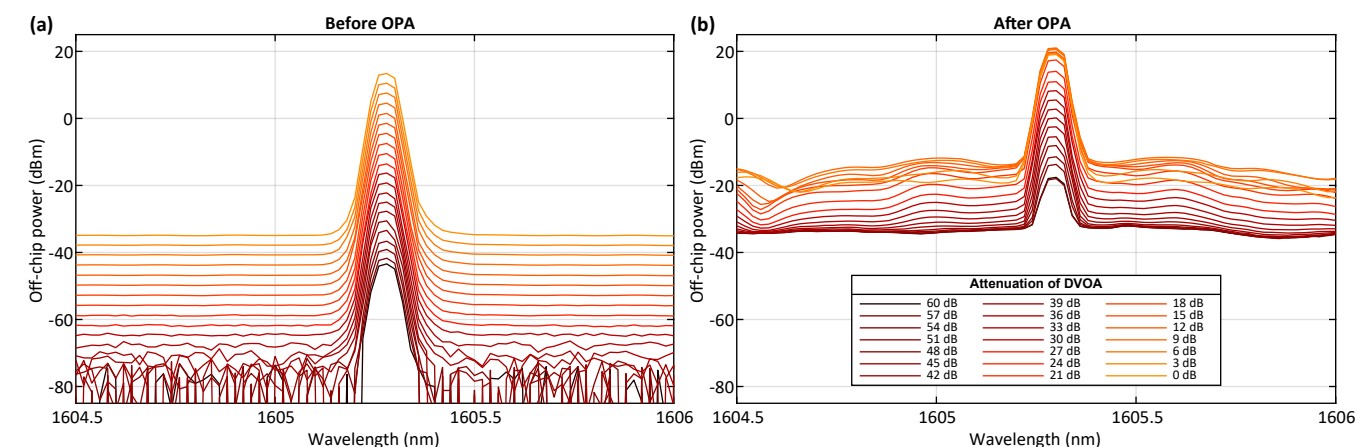

**Extended Data Fig. 8 | Amplification measurements with varying input signal power. a**, Optical spectra before amplification. The signal wavelength is chosen within the region of good transmission at the maximum of the parametric gain lobe to achieve the highest gain. The input power is swept from 46 nW (−43.4 dBm) to 21.9 mW (13.4 dBm) in steps of 3 dB. The optical SNR of the input signal exceeds 48 dB. **b**, Optical spectra after amplification. An increased noise floor owing to the parametric fluorescence is observed. As attenuation approaches 0 dB, the peak power exhibits only small changes, indicating amplifier saturation.

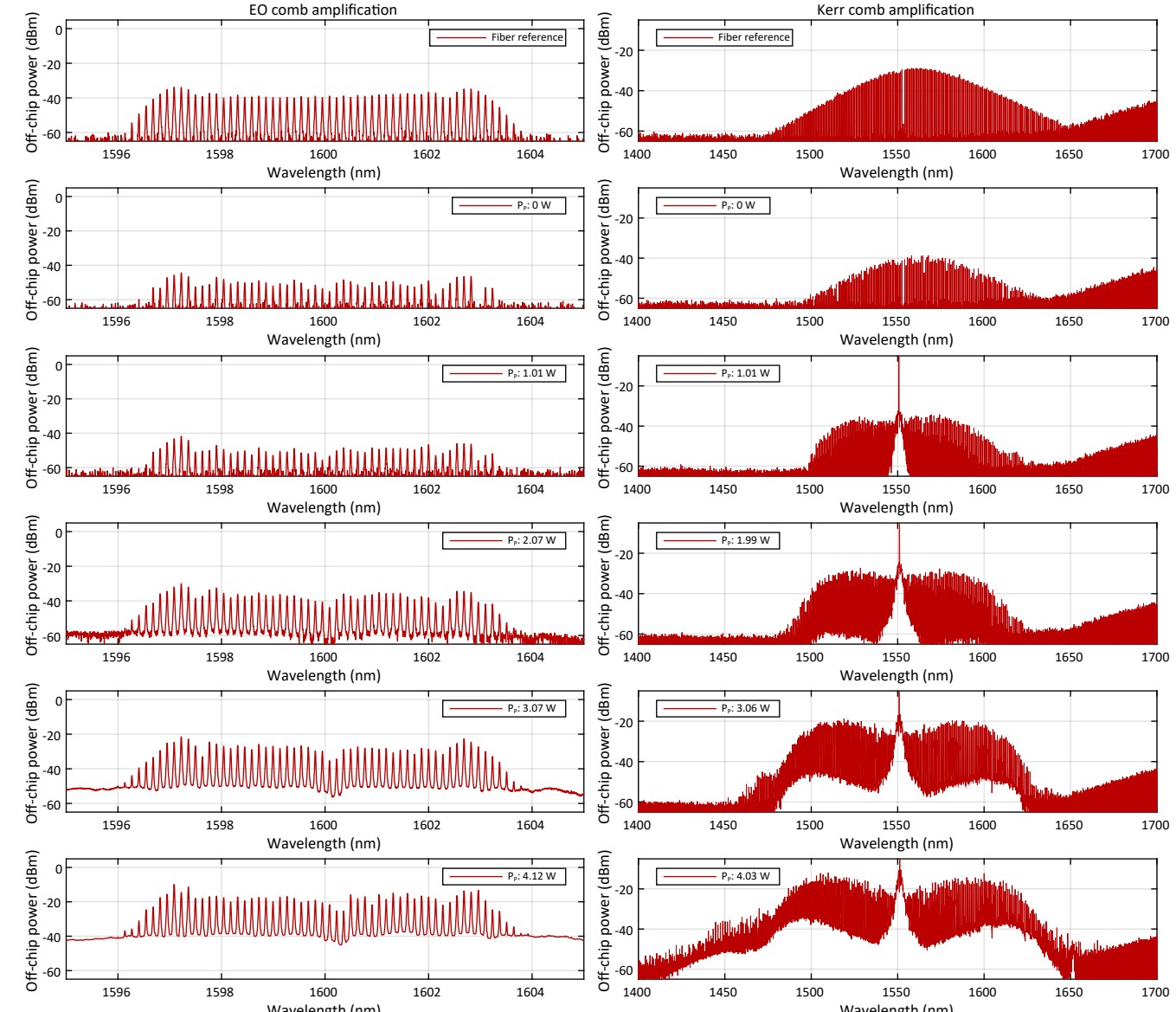

**Extended Data Fig. 9 | Extended amplification measurement datasets for the EO comb and Kerr comb.** Here $P_P$ is an off-chip pump power. Left column, EO comb amplification. The amplification is relatively uniform at the peak of the parametric gain lobe, with each line of the EO comb being amplified by approximately the same amount. The dip in the spectrum at wavelengths slightly above 1,600 nm is caused by an increased loss in this spectral region, probably because of a defect in the waveguide or interference with higher-order modes. Right column, Kerr comb amplification. The enhanced signal comb overlaps the generated idler comb. At the highest pump power, two distinct spectral features can be observed: more lines from non-degenerate FWM and a small peak around 1,650 nm attributed to the Raman effect. The elevated noise floor at longer wavelengths is because of the roll-off of the OSA used to measure the spectra.