## [Peer Review File · Nature]

An ultra-broadband photonic-chip-based parametric amplifier

Corresponding Author: Professor Tobias Kippenberg

Version 0:

Reviewer comments:

Referee #1

(Remarks to the Author)

The paper presents an ultra-broadband on-chip optical amplifier, which is useful and promising. The paper overall is very well written. In my opinion, the advances in the technology that are presented in this paper are worth publishing. I have the following comments and believe they need to be addressed.

1. My main concern is that though the amplifier seems to be primarily developed to address telecommunication applications, the allowed input power (or the saturation of the amplifier) seems to be very low. This is the most obvious from the section 'amplification of optical frequency combs'. Both the EO comb and the soliton comb were kept at a very low power sending into the amplifier. For instance, the EO comb is ~ -50 dBm total power which is extremely low for a typical EO comb. In fact, it is probably difficult to generate an EO comb with that little power without manually/voluntarily attenuating it. A typical total power of an E/O comb is in the range of several dBm, and that would put the amplifier in saturation by observing Fig.2c. The soliton comb is in a very similar situation (~ -41 dBm).

Natural questions from the above observation would be i) whether the amplifier is still relevant for telecommunications, especially the scenario the paper is describing (amplifying combs); ii) what limits the saturation power and what are the potential ways to overcome it, if there may be a way.

2. Figure 2 c,d,e, are a bit confusing to me. If observing one of these three figures, and comparing the curves for different pump powers, why the gain can be > 10 dB higher while the pump power only increases by 3 dB (e.g. from 2 W to 4 W)?

3. Figure 1b does not have values for the vertical axis. The values are necessary, otherwise little knowledge/information can be gained from the whole plot.

4. It is explained that the 9% conversion efficiency is limited by the waveguide propagation loss, but I don't seem to find the actual propagation loss in the paper (only a number that summarizes coupling and propagation which is not as helpful).

5. The coherent data transmission setup needs to be described in more details. There is no information about the coherent receiver (e.g. photodiode bandwidth or model, if there are electrical amplifiers after the PDs, what is the model/bandwidth of the ADC, and if there is digital signal processing applied to recover the signals (if yes, the steps/blocks of DSP need to be described)). This information is very helpful for readers to understand the constellations shown in Fig.4 c,d,e. The AWG used in the setup can create constellations with much higher quality than the shown constellations. It is common that there are degradations from the electrical to optical domain, it is important to understand what is done at the receiver so better knowledge can be gained for the amplifier in the paper. Also the optical power (LO and signal) going into the coherent needs to be mentioned.

6. This is a minor comment. It is very uncommon to use 'MER' (assuming modulation error ratio. Didn't manage to find the explanation in the paper). Signal-to-noise ratio (SNR) is much more common. A SNR in this context would mean constellation SNR (or sometimes referred to as 'symbol SNR'), which should be the same definition as the MER if I understand the MER in the paper correctly. But MER is very uncommon.

Referee #2

(Remarks to the Author)

This article presents the demonstration of an integrated optical parametric amplifier offering a wide bandwidth of 140 nm with

a net gain of 10 dB. This performance was achieved using a highly non-linear material, gallium phosphide dioxide on silicon. The high non-linearity enables to use a shorter length, thus limiting the negative impact of linear losses on the process. In addition, it requires lower pump power than other materials. The authors have characterized the gain, noise figure and frequency conversion properties of the amplifiers. They have also demonstrated their ability to amplify broadband signals such as frequency combs and high speed telecommunication signals.

The article is well written and illustrated. The importance of this work is clearly detailed, and its positioning in relation to the state of the art is well argued. The performance is impressive, particularly in terms of bandwidth, exceeding that of standard EDFA systems. This work demonstrates the promising potential applications of these integrated parametric amplifiers in future telecommunications systems.

However, I have two major problems with the publication of this paper in Nature. Firstly, similar work was published in Nature two years ago (Ref 11, entitled "A photonic integrated continuous-travelling-wave parametric amplifier"), albeit with a different material and a narrower gain bandwidth. Secondly, there is a significant discrepancy between numerical predictions and experimental results (see Extended Data Fig. 2). The pump power required is more than double the predicted value, and the observed gain bandwidth is 140 nm when it should be 500 nm. This suggests that further study is required before publication. The question is therefore whether two papers on the same subject can be published in Nature by partly similar authors, while the second shows a considerable improvement in performance thanks to the use of a new material. The second point is technical, although it has raised serious concerns. My recommendation would be to reject the paper and to submit it to a more specialized journal.

In addition, the following comments should be considered before publication:

1. The term "traveling-wave parametric amplifier" can be confusing for readers. If this is an integrated waveguide parametric amplifier based on the same physical process as the first fiber parametric amplifiers, it would be clearer to retain this terminology. Otherwise, the authors should clearly explain the differences from the work of M. Marhic and P. Andrekson on fibers, for example.
2. The authors claim that these amplifiers are unidirectional, which is true. However, it is unlikely that the isolator in the EDFA (used to achieve 4 W at the input) can be removed. This assertion should therefore be reconsidered.
3. When we talk about input power, do we mean the power injected into the guide or at the output of the EDFA?
4. The dispersion of the waveguide is not measured. Can it be provided? This is crucial since the β_2 and β_4 values are used as fitting parameters for the simulations to try and get agreement with the experimental results. What is the β_3 value? This question is important because significant disagreement between simulations and experiments (more than double the required pumping power and a different gain shape) could indicate incorrect values or non-uniform waveguides. Further information on these points is required. The literature on fiber-based systems is abundant on this subject (see Marhic's book and related references).
5. The statement "in principle, the pump frequency can be freely chosen over a wide frequency range" is inaccurate. Changing the pump wavelength significantly alters the parametric gain shape, except when $\beta_3=0$.
6. The SBS gain in these waveguides is low. An estimate of the power threshold would be useful to determine the maximum pump power achievable with phase modulation schemes on the pump.
7. There is an inconsistency in the net gain values: 10 dB are mentioned, but 25 dB are noted in the discussion. Clarification is needed.
8. Although these results are impressive, the authors' anticipation of the use of these amplifiers as "ideal candidates for driving next-generation fiber systems based on doubly anti-resonant hollow-core fibers" seems premature. Given the strong non-linearity of the waveguide, strong FWM interactions between the signals are expected and need to be verified before such claims can be made.

Version 1:

Reviewer comments:

Referee #1

(Remarks to the Author)

The authors have answered all my questions very well, and made changes to the paper accordingly. Technically, I don't agree 100% with the authors but I do believe the quality of the paper meets the publication level. The paper should be published in my opinion.

Referee #2

(Remarks to the Author)

Dear editor, I am satisfied with most of the authors' responses and recommend the publication of the paper. However, I remain unconvinced regarding the use of the term "traveling wave amplifier." The authors argue that "it is more important to highlight the physics of the process rather than the platform," but this is precisely my point. These amplifiers rely on phase-matched four-wave mixing processes in $\chi(3)$ media, regardless of the platform. Therefore, I maintain that this is more accurately described as a "parametric amplifier," which would be more familiar and clear to the optics community focused on this field. Referring to it simply as a "parametric amplifier" would not detract from the novelty and would enhance clarity.

Submission of a revised version of our manuscript “An ultra-broadband photonic-chip-based traveling-wave parametric amplifier”

We thank the referees for the detailed review and suggestions to improve the manuscript. Here we present a point-by-point response to the review. The original text of the report is printed in black, our replies are highlighted in blue, and the action taken in red.

Summary of reply to reviewers:

Referee #1 (Remarks to the Author):

The paper presents an ultra-broadband on-chip optical amplifier, which is useful and promising. The paper overall is very well written. In my opinion, the advances in the technology that are presented in this paper are worth publishing.

We thank the referee for the effort to review our manuscript and appreciate that the reviewer recognizes the utility and promise of the reported results on broadband amplification using an integrated TWPA based on GaP.

I have the following comments and believe they need to be addressed.

1. My main concern is that though the amplifier seems to be primarily developed to address telecommunication applications, the allowed input power (or the saturation of the amplifier) seems to be very low. This is the most obvious from the section ‘amplification of optical frequency combs’. Both the EO comb and the soliton comb were kept at a very low power sending into the amplifier. For instance, the EO comb is ~ -50 dBm total power which is extremely low for a typical EO comb. In fact, it is probably difficult to generate an EO comb with that little power without manually/voluntarily attenuating it. A typical total power of an E/O comb is in the range of several dBm, and that would put the amplifier in saturation by observing Fig.2c. The soliton comb is in a very similar situation (~ -41 dBm).

We would like to clarify that while we have emphasized indeed the application in optical telecommunications, it is not our intention to single out this application, nor are we certain given the stringent demands of optical communication technology. As such, we have revised our manuscript to note the range of applications in which amplifiers with broadband, high gain, and a large dynamic range can be utilized.

Concerning the specific comment on the input power/saturation being very low: We agree that input/output powers are important characteristics of optical amplifiers. We respectfully note however that the values mentioned in the review **do not correspond** to those stated in our manuscript. The total off-chip power of the EO comb injected into the chip (i.e., after a 10 dB signal loss in the fiber splitter used to combine the pump and signal waves) is $10.5 \mu\text{W}$, as written on line 172. That corresponds to -19.8 dBm, while the power per line varies between -35 dBm and -40 dBm, as shown in Fig. 3(b). After amplification, the power per line varies between roughly -15 and -20 dBm. The same applies to the soliton microcomb; the total off-chip input power stated in the manuscript is $76 \mu\text{W}$ (line 179), which is equivalent to -11.2 dBm with a power per line up to -30 dBm. These total power values differ from the values in the comment above by roughly 30 dB. For this reason, the amplifier could be utilized also in an optical communication setting.

Action taken: We have highlighted the range of potential applications beyond telecommunications in the Discussion section.

Natural questions from the above observation would be i) whether the amplifier is still relevant for telecommunications, especially the scenario the paper is describing (amplifying combs); ii) what limits the saturation power and what are the potential ways to overcome it, if there may be a way.

i) Given the explanations above, we do believe that the amplifier is relevant for telecommunications (and we focus in our manuscript therefore on the proof of concept example of applying the weak lines of a microcomb that require significant post amplification to be used as carriers in a parallel coherent communication transmission system), as it operates within or close to the power ranges required for the optical transceivers and receivers used in telecommunication technology – for example, from -8 dBm to +6 dBm for transceivers and from -26 dBm to 0 dBm for coherent receivers, as defined in the Recommendation ITU-T G.698.2. In Fig. 2(f) we demonstrate that the effect of saturation becomes evident for the single input line located at the maximum of the gain lobe only when the input power approaches -5 dBm. The off-chip output power at saturation in this case is 21 dBm (125 mW).

A rough estimation can be made of the implications of using the amplifier to boost the input of a frequency comb for parallel coherent communications: for 100 lines, saturation will occur only if the input power of each line reaches -25 dBm and the gain reaches 25 dB, in which case the output would be 100 lines at the level of 0 dBm, or 20 dBm of total power. In practice, the gain is lower outside of the gain lobes, and the input power can be even higher. These power levels are well within the typical values used in telecommunications.

Regarding the amplification of combs, it should be noted that the input power of our combs was limited by the low efficiency of Kerr comb generation, the high insertion loss of the EO phase modulators used for EO comb generation, and the additional 10 dB of attenuation of the signal by the 90/10 fiber splitter that we used to combine signal and pump before the chip. The latter loss can be avoided in future experiments by using wavelength-division multiplexers.

ii) The saturation power is limited by the available pump power that can be transferred to the signal during the amplification and the optical propagation loss of the waveguide. To extend the limits, it is required to further advance the fabrication methods to reduce propagation losses so more energy can be transferred to the signal instead of being lost; reduced propagation losses are key and would help to enhance the gain and/or gain bandwidth depending on the length of the waveguide. The absolute conversion efficiency is ultimately limited to 50% due to the generation of the idler.

In conclusion, we would like to offer a few broader observations beyond the specific points addressed. The amplifier we discuss presents several significant advantages, applicable not only in optical communications but also across various fields. One key benefit is its wide dynamic range, enabling the amplification of even very weak signals (we show in this context 6 orders of magnitude, i.e. 60 dB). Achieving this with an EDFA would typically require specialized, low-doping amplifiers, as high doping levels would cause ASE noise to dominate, resulting in a fixed minimum input power for an acceptable noise figure. In contrast, the TWPA offers a noise figure that remains almost independent of the input signal power level, which is a key advantage for any application that has a low input light power level (e.g. LiDAR). Finally, while not covered in depth here, the nature of the amplification process allows for applications such as spectral cloning and the mitigation of signal impairments. Indeed, there is a rich body of literature that has shown the unique properties that phase-sensitive amplification can offer, also concerning the ‘passive’ elimination of signal transmission impairments.

Last, a further unique aspect of the TWPA amplifier lies in its ability to provide broadband gain and gain in regions that are not covered by rare earth ions. For example, TWPA can provide gain also at 1.3 μm , which is used in data-center pluggable optics, or 1650 nm which is a key absorption line for Methane. To date, the only methods to amplify in the 1.3 μm and 1.6 μm spectral range are III-V amplifiers or fibers doped with Bismuth for 1.3. Here the TWPA approach is a new, admittedly yet in its infancy, technology.

Finally, from an integrated photonics perspective, the TWPA are attractive as they are more resilient to backreflection from the chip facets. Since the gain is unidirectional, the sensitivity to reflection is not linear, but second order in reflection. Therefore, TWPA can support high gain, without entering the mode of spurious lasing (which e.g. can be the case for III-V amplifier dies, that generally do not support such large gain of ca. 30 dB when integrated on a silicon die).

Action taken: A statement has been added to the discussion section highlighting that our amplifier covers the power range required for telecommunications.

2. Figure 2 c,d,e, are a bit confusing to me. If observing one of these three figures, and comparing the curves for different pump powers, why the gain can be > 10 dB higher while the pump power only increases by 3 dB (e.g. from 2 W to 4 W)?

We thank the reviewer for the question, which hints at one intriguing property of this amplifier. The dependence of the Kerr parametric gain on the pump power can be explained using standard coupled-mode theory that is described, for example, in [1]. The maximum gain in the small-signal regime scales with the pump power as $G_s = 1 + [\sinh(\gamma P_p L)]^2 \approx \frac{1}{4} \exp(2\gamma P_p L)$, where γ is the effective nonlinearity, P_p is the pump power and L is the waveguide length (or effective length if the attenuation in the waveguide is not negligible). This results in exponential scaling of the gain with a linear change of the pump power, or in dB units, a linear dependence of the gain on the pump power in W. Closer to the pump wavelength, the dependence smoothly changes to quadratic becoming $G_s \approx (\gamma P_p L)^2$ as a result of imperfect nonlinear phase matching. The gain scales more slowly only if the amplifier is saturated or other processes such as nonlinear absorption or interference with higher-order modes are present, depleting a portion of the pump power. Thus, if the pump power is increased by 3 dB, in a lossless system one would always observe an increase in the gain by at least 6 dB, and a significantly stronger gain increment is observed for optical frequencies satisfying the zero phase-mismatch condition (around the maximum of the so-called “parametric gain lobes”). If losses are present, the scaling is slower but is still nonlinear. This is in good agreement with the results of the numerical simulations presented in our manuscript (see Extended Figure 2), and this explains the scaling of the parametric gain in our measurements.

Action taken: A sentence on the exponential gain scaling has been added to the main manuscript.

[1] Hansryd, J., Andrekson, P. A., Westlund, M., Li, J. & Hedekvist, P.-O. Fiber-based optical parametric amplifiers and their applications. *IEEE Journal of Selected Topics in Quantum Electronics* 8, 506–520 (2002).

3. Figure 1b does not have values for the vertical axis. The values are necessary, otherwise little knowledge/information can be gained from the whole plot.

We thank the referee for catching this embarrassing mistake. The idea behind this graph is to compare only the bandwidth of different amplifiers by plotting normalized gain, but we, unfortunately, neglected to indicate the magnitude of the vertical divisions.

Action taken: Figure 1b has been updated with the magnitude of the divisions on the vertical axis.

4. It is explained that the 9% conversion efficiency is limited by the waveguide propagation loss, but I don't seem to find the actual propagation loss in the paper (only a number that summarizes coupling and propagation which is not as helpful).

The transmission characterization and loss measurements are described in the corresponding subsection in Methods. In particular, in line 546 of the originally submitted version it is written "the average loss rate is equal to 0.8 dB cm⁻¹ (Extended Data Fig. 3(b))." We agree that this is an important parameter that should be mentioned in the main text.

Action taken: The actual propagation loss value has been added to the main text (subsection 'Optical gain measurements').

5. The coherent data transmission setup needs to be described in more details. There is no information about the coherent receiver (e.g. photodiode bandwidth or model, if there are electrical amplifiers after the PDs, what is the model/bandwidth of the ADC, and if there is digital signal processing applied to recover the signals (if yes, the steps/blocks of DSP need to be described)). This information is very helpful for readers to understand the constellations shown in Fig.4 c,d,e. The AWG used in the setup can create constellations with much higher quality than the shown constellations. It is common that there are degradations from the electrical to optical domain, it is important to understand what is done at the receiver so better knowledge can be gained for the amplifier in the paper. Also the optical power (LO and signal) going into the coherent needs to be mentioned.

We thank the reviewer for the comments on the coherent data transmission setup and for spotting the omitted LO power values. We agree that it would be very helpful for readers to know more details about the setup in addition to our description presented in the subsection "Coherent communication" in Methods. We indeed applied digital signal processing, and the description of steps is now also included in Methods. The codes used to process the data will be freely available upon publication, as declared in the manuscript.

Action taken: Further details of the data transmission setup have been provided in the Methods section, including model numbers for key equipment and a step-by-step explanation of the digital signal processing procedures. We added the missing value of the LO power to the main text.

6. This is a minor comment. It is very uncommon to use 'MER' (assuming modulation error ratio. Didn't manage to find the explanation in the paper). Signal-to-noise ratio (SNR) is much more common. A SNR in this context would mean constellation SNR (or sometimes referred to as 'symbol SNR'), which should be the same definition as the MER if I understand the MER in the paper correctly. But MER is very uncommon.

The definition of the constellation SNR, or the symbol SNR, is indeed very similar to the definition of the modulation error ratio (MER) that we use in our work. We define MER in line 214 of the originally submitted version of our manuscript. The definition we use is the standard definition of this figure of merit [2, 3], identical to the definition of the function in the Communication Toolbox of MATLAB that we used to digitally process the data:

$$\text{MER} = 10 \log_{10} \left(\frac{\frac{1}{N} \sum_{n=1}^N [(I_{\text{ideal},n})^2 + (Q_{\text{ideal},n})^2]}{\frac{1}{N} \sum_{n=1}^N [(I_{\text{received},n} - I_{\text{ideal},n})^2 + (Q_{\text{received},n} - Q_{\text{ideal},n})^2]} \right).$$

However, there is still a small difference between MER and constellation SNR: when SNR is calculated, the average power of actually measured symbol vectors is considered, and the power of error vectors is

evaluated with respect to the average signal power. However, for the calculation of MER, the average power of ideal symbol vectors is used, and the power of error vectors is calculated with respect to the ideal constellation point. Therefore, SNR and MER are equivalent if only Gaussian noise is present in the system. In this case, the mean value of the actual received data and the ideal constellation point are the same. MER can be more representative because it accounts for all types of distortions and errors, not just noise, including the distortions caused by amplifiers, in particular when using nonlinear optical amplifiers. Since the use of MER is somewhat uncommon in the field, we have extended the discussion of the parameter definition and its relationship to SNR.

[2] ESTI TR 101 290. "Digital Video Broadcasting (DVB): Measurement guidelines for DVB systems." June 2020

[3] Recommendation ITU-R BT.1735-3

Action taken: the definition of MER is explicitly added as an equation to Methods; an explanation is added to confirm that MER and constellation SNR are equivalent to each other for our experimental data.

Referee #2 (Remarks to the Author):

This article presents the demonstration of an integrated optical parametric amplifier offering a wide bandwidth of 140 nm with a net gain of 10 dB. This performance was achieved using a highly non-linear material, gallium phosphide dioxide on silicon. The high non-linearity enables to use a shorter length, thus limiting the negative impact of linear losses on the process. In addition, it requires lower pump power than other materials. The authors have characterized the gain, noise figure and frequency conversion properties of the amplifiers. They have also demonstrated their ability to amplify broadband signals such as frequency combs and high speed telecommunication signals.

The article is well written and illustrated. The importance of this work is clearly detailed, and its positioning in relation to the state of the art is well argued. The performance is impressive, particularly in terms of bandwidth, exceeding that of standard EDFA systems. This work demonstrates the promising potential applications of these integrated parametric amplifiers in future telecommunications systems.

We would like to thank the referee for the appreciation of our work and our results.

However, I have two major problems with the publication of this paper in Nature. Firstly, similar work was published in Nature two years ago (Ref 11, entitled "A photonic integrated continuous-travelling-wave parametric amplifier"), albeit with a different material and a narrower gain bandwidth.

We thank the reviewer for raising these important questions, and we are glad to have been given the opportunity to clarify them. The key novelty of the current work is that using high-quality silica-cladded gallium phosphide waveguides, we have developed a photonic chip-based parametric amplifier that operates in a completely new regime – a regime that is highly relevant for the potential adoption of nonlinear amplification technologies in photonics. Compared to the referenced work in silicon nitride, the GaP platform simultaneously provides multiple qualitative breakthroughs:

1. **Ultra-broad bandwidth:** The bandwidth covers 140 nm, compared to the bandwidth of less than 20 nm in Ref. 11 and nearly three times the bandwidth of an erbium amplifier, and is quantitatively a new regime.
2. **High gain:** The **off-chip** net parametric gain of the GaP OPA reaches a maximum value of **25 dB** and exceeds 10 dB over the entire bandwidth. In the previous first-ever net gain result, we achieved

off-chip net gain (a maximum of only 2 dB), requiring almost twice the highest pump power (6.7 W) used here.

3. **Ultra-compact footprint:** The waveguide length of the new amplifier is just 5.55 cm, which is more than 35 times shorter and occupies a 60-times smaller chip area than the 2-meter-long waveguides in Ref. 11.

We also demonstrate the parametric amplification of extremely low signals, down to the nW level, and we show that the dynamic range of amplifiable input signals exceeds six orders of magnitude. Moreover, the conversion efficiency is sufficiently strong that inter-band communication signal translation from the signal to the idler (from L-band to S-band) can be measured directly without any post-amplification of the idler. Materials like silicon, chalcogenides, Si₇N₃, AlGaAs, and many others have so far been unable to produce strong CW gain due to two-photon absorption and high propagation losses, while GaP has negligible two-photon absorption at telecom wavelengths and an acceptable level of propagation losses.

Secondly, there is a significant discrepancy between numerical predictions and experimental results (see Extended Data Fig. 2). The pump power required is more than double the predicted value, and the observed gain bandwidth is 140 nm when it should be 500 nm. This suggests that further study is required before publication. The question is therefore whether two papers on the same subject can be published in Nature by partly similar authors, while the second shows a considerable improvement in performance thanks to the use of a new material. The second point is technical, although it has raised serious concerns. My recommendation would be to reject the paper and to submit it to a more specialized journal.

Concerning the discrepancy between the numerical predictions and experimental results, we respectfully point out that we present a detailed discussion in the subsection “Numerical calculations of optical parametric gain” in Methods. Numerical simulations are based on simple coupled-mode equations that do not take into account some nonlinear phenomena, such as nonlinear absorption and device imperfections. The waveguide dispersion is designed using FEM calculations, and the dimensions of the real waveguide as well as the material refractive index models can slightly differ from measured or literature values. Uncertainty in the material nonlinearity, the presence of higher-order modes, frequency-dependent nonlinearity, and the effect of three-photon absorption can affect the efficiency of the four-wave mixing process, leading to more optimistic values of pump power in the numerical simulations. This topic remains an on-going field of research for waveguide parametric amplifiers.

It should also be noted that the β_2 parameter is extremely sensitive to the waveguide dimensions, especially if we would like to work close to the zero-dispersion point; the tolerance is only a few nm. In fact, none of the established photonic platforms are able to achieve that level of precision to our knowledge, which is why typically multiple devices with slightly varied geometries are fabricated, or devices are designed to accommodate statistical discrepancies. Given the limited space on the chip and the time-consuming fabrication, we did not have enough statistical data to implement a possible geometry correction to compensate for fabrication deviations. Thus, the difference in the designed and measured bandwidth is due to the relative novelty of the fabrication routines and stringent tolerances. We are currently updating our simulation routines to be able to better predict the performance of parametric amplifiers in our future work.

Action taken: The section on numerical calculations of optical parametric gain has been extended to include a discussion of the fabrication tolerance and a corresponding figure.

1. The term "traveling-wave parametric amplifier" can be confusing for readers. If this is an integrated waveguide parametric amplifier based on the same physical process as the first fiber parametric amplifiers, it would be clearer to retain this terminology. Otherwise, the authors should clearly explain the differences from the work of M. Marhic and P. Andrekson on fibers, for example.

Indeed, our parametric amplifier is based on the same physical process as the first fiber parametric amplifiers, as in the work of M. Marhic and P. Andrekson. It is not our intention to confuse or obfuscate the origin of our work, which we have clearly highlighted in our manuscript in the opening paragraph. The term “traveling-wave parametric amplifier”, or TWPA, originates from the field of microwave engineering before fiber-based amplifiers were developed, as the referee will be aware of. Evidently, it refers to a type of amplifier that utilizes the propagation of electromagnetic waves along a transmission line or waveguide to achieve amplification, rather than resonant structures that have also been employed. As such the key concept behind a traveling-wave amplifier is the interaction between electromagnetic waves traveling in the same direction. Moreover, this term has in fact also been used (perhaps less frequently) in the context of fiber amplifiers – for example, S.-K. Choi, R.-D. Li, C. Kim, and P. Kumar, “Traveling-wave optical parametric amplifier: Investigation of its phase-sensitive and phase-insensitive gain response,” J. Opt. Soc. Amer. B, vol. 14, pp. 1564–1575, 1997. It is noted that microwave TWPA are in fact still a very active research field – in particular those based on Josephson Junctions (and surprisingly, as a side note, much of the literature of fiber parametric amplifiers seems to be entirely unknown to this community. Our work therefore also assists in bridging a link between communities as the concepts used are evidently identical. As a further side remark: ironically the concept of phase matching using waveguide dispersion was only recently realized in the microwave TWPA community!).

One more reason we would like to refrain from using an “integrated waveguide” parametric amplifier is that this term has not been used in the community, while the “parametric gain” is evidently used in microresonators for the purpose of frequency comb generation, and more recently also in works that build ‘resonant’ parametric amplifiers (that also refer to parametric amplifiers, but do not specify the fact that they are resonant) [4].

For this reason, however, we would like to keep using the conventional term “traveling-wave” since, in our opinion, it is more important to highlight the physics of the process rather than the platform on which the amplifier is implemented.

[4] Zhao Y, Jang JK, Ji X, Okawachi Y, Lipson M, Gaeta AL. Large regenerative parametric amplification on chip at ultra-low pump powers. *Optica*. 2023 Jul 20;10(7):819-25.

Action taken: To clarify the nature of our amplifier, we stated in the introductory paragraph that this device is equivalent to a fiber parametric amplifier but uses integrated waveguides.

2. The authors claim that these amplifiers are unidirectional, which is true. However, it is unlikely that the isolator in the EDFA (used to achieve 4 W at the input) can be removed. This assertion should therefore be reconsidered.

We thank the referee for the valuable comment. Indeed, the isolator in the EDFA cannot be removed at this time. We refer in the text on the reduced requirement for isolators before and after the parametric amplifier in the signal path. This is particularly true for links reliant on cascaded amplification chains. In the future, given modest improvements in GaP waveguide loss and semiconductor single-mode laser diode output power, which today is as great as 500 mW at 1550 nm, it is conceivable that the amplifier could be operated without an EDFA at a pump power of less than 1 W. Linking back to the comparison with Si₃N₄ waveguides, where a CW OPA was only achieved at the culmination of a ten-year effort in nanofabrication process optimization, this point serves to further highlight the great potential of GaP for the future of TWPA systems.

We would however like to also comment that one key advantage of the unidirectional nature is the insensitivity to reflection from the chip facet. In a semiconductor or Er-based integrated waveguide

amplifier, the reflection sensitivity with respect to the gain is linear in reflection, while quadratic in the current case (as the gain is only in one direction). This implies parasitic lasing is significantly suppressed, which can readily occur with high gain on a chip.

Action taken: The assertion about isolators has been updated to mention the isolator of the pump EDFA.

3. When we talk about input power, do we mean the power injected into the guide or at the output of the EDFA?

As stated in lines 559 - 561 of the original version of the manuscript, all powers in the figures and in the text indicate the values at the tips of the input or output lensed fibers tips, i.e. off-chip. We prefer this definition of pump power, as GaP is a passive optical material at telecom wavelengths, and some coupling loss from the pump source to the waveguide cannot be avoided, even accounting for further integration of the pump source. Since we refer to the off-chip (fiber tip to fiber tip) net gain as the most important performance characteristic of our OPA, we believe that the pump power at the input fiber is the best corresponding measurement point. Improvements in optical coupling losses may decrease the pump power substantially in the future.

As an example, given the coupling losses of 2.5 dB/facet, 4 W of pump power at the tip of the lensed fiber (as used in the manuscript) corresponds to ~ 2.24 W of pump power at the beginning of the waveguide, and this corresponds to ~ 9 W directly at the output fiber of the EDFA after the internal isolator.

4. The dispersion of the waveguide is not measured. Can it be provided? This is crucial since the β_2 and β_4 values are used as fitting parameters for the simulations to try and get agreement with the experimental results. What is the β_3 value? This question is important because significant disagreement between simulations and experiments (more than double the required pumping power and a different gain shape) could indicate incorrect values or non-uniform waveguides. Further information on these points is required. The literature on fiber-based systems is abundant on this subject (see Marhic's book and related references).

We would like to respectfully point out that the dispersion measurements are explained and provided in the subsection "Numerical calculations of optical parametric gain" in the Methods section, starting at line 485 of the original manuscript. The experimentally measured value of β_2 is $-124 \text{ fs}^2\text{mm}^{-1}$ and we do not provide values for β_3 and β_4 because the uncertainty of our measurements is too large due to the short length of the GaP waveguide. We also do not observe the 4th-order dispersion gain lobes in our measurements.

As described above, the requirements for the waveguide dimensions are extremely stringent to attain the 4th order phase matching that maximizes the gain bandwidth. Even a few nanometers of discrepancy cause significant changes in the dispersion profile.

Action taken: The dispersion fit plot has been added to Methods, and the experimentally measured second-order dispersion value has been added to the main text.

5. The statement "in principle, the pump frequency can be freely chosen over a wide frequency range" is inaccurate. Changing the pump wavelength significantly alters the parametric gain shape, except when $\beta_3=0$.

We apologize for the confusion. Indeed, changing the pump wavelength significantly alters the parametric gain shape for the amplifier that is already fabricated. What we meant is that one can freely choose the pump wavelength and then design the amplifier waveguide cross-section accordingly. Different amplification wavelengths may be collocated on the same GaP chip. Ultimately, the range of pump frequencies is limited only by the transparency window and the nonlinear absorption threshold wavelengths of the material, unlike in rare-earth-doped amplifiers, for example.

Action taken: We have added a statement clarifying that the pump wavelength can be chosen freely during the design process.

6. The SBS gain in these waveguides is low. An estimate of the power threshold would be useful to determine the maximum pump power achievable with phase modulation schemes on the pump.

While the Kerr effect is generally enhanced in nonlinear waveguides compared to optical fibers, the SBS gain is indeed known to be lower, for example, in Si_3N_4 [5]. To our knowledge, the SBS gain has not been explicitly studied in integrated GaP waveguides, and careful investigation of this is the subject of future experimental work.

The Brillouin gain can be calculated as given in [5]: $g_B = \frac{4\pi^2 n^7 p_{12}^2 \eta}{\lambda^2 c \rho v_a \Gamma}$, where n is the material refractive index, c is the speed of light in vacuum, ρ is the material density, p_{12} is the photoelastic constant, Γ is the acoustic damping, v_a is the longitudinal sound velocity and η is the acousto-optic overlap coefficient. The refractive index of GaP at a vacuum wavelength of 1550 nm is 3.05, which is roughly 1.53 times larger than the refractive index of Si_3N_4 . The photoelastic constant of GaP is 2.76 times larger and the material density of GaP is 1.31 times larger than the corresponding values for Si_3N_4 [6]. These parameters taken together would contribute an approximately 114-fold enhancement to the Brillouin gain in GaP. However, GaP is not an amorphous material, and the longitudinal sound velocity, which is about half that in Si_3N_4 , depends significantly on the direction in the crystal, an important consideration for spiral waveguides. Moreover, the acoustic damping is expected to be significantly elevated due to the waveguide nonuniformity and increased phonon leakage to the substrate, but the exact value can be obtained only through a dedicated experiment, which is out of the scope of our work. The acousto-optic overlap coefficient should not differ significantly from the value used in [5], as the waveguide geometry is similar, and even in the case of perfect overlap, the Brillouin gain would increase by at most a factor of 4 compared to the

waveguide in [5]. Therefore, if we ignore our assumption about the elevated acoustic damping to avoid underestimation and assume that the acoustic damping in GaP spiral waveguides is not significantly different from Si₃N₄, we can expect the Brillouin gain in GaP devices to be at most 230 times higher than in [5], or in other words, still almost 1.5 times less than Brillouin gain in single-mode fibers, but not higher. A typical value for the Brillouin threshold for a 20-km-long single-mode fiber at 1550 nm is 1 mW [7]. The maximum effective length of our GaP waveguides does not exceed 3.5 cm, but the effective area is almost 200 times smaller. Thus, we can estimate that the Brillouin threshold in GaP spirals to be approximately 4 W. This is close to the power at the fiber tip in the highest power experiments. Even with our overestimated value, calculated using an acousto-optic overlap coefficient of 1, we can conclude that one should not expect a strong Brillouin scattering in GaP waveguides, and there is no need for the phase modulation of the pump as evidenced by our measurements with a low linewidth pump laser. One can expect lower Brillouin gain due to an acousto-optic overlap coefficient of less than 1, but the acoustic damping remains unknown. In our experiments, we did not observe any additional pump power depletion even at the highest power levels. As we noted, for off-chip pump power of 4.5 W, a parametric comb is spontaneously formed (described in the Methods) due to the strong parametric gain and reflections on the chip facets. So far, this is the limiting factor for our amplifiers and appears before any evidence of SBS. Therefore, we leave the investigation of the Brillouin effect for future work, assuming for now that it is negligible.

[5] Gyger F, Liu J, Yang F, He J, Raja AS, Wang RN, Bhave SA, Kippenberg TJ, Thévenaz L. Observation of stimulated Brillouin scattering in silicon nitride integrated waveguides. *Physical review letters*. 2020 Jan 10;124(1):013902.

[6] Schneider K, Baumgartner Y, Hönl S, Welter P, Hahn H, Wilson DJ, Czornomaz L, Seidler P. Optomechanics with one-dimensional gallium phosphide photonic crystal cavities. *Optica*. 2019 May 20;6(5):577-84.

[7] Agrawal GP. Nonlinear fiber optics. In *Nonlinear Science at the Dawn of the 21st Century* 2000 Dec 21 (pp. 195-211). Berlin, Heidelberg: Springer Berlin Heidelberg.

7. There is an inconsistency in the net gain values: 10 dB are mentioned, but 25 dB are noted in the discussion. Clarification is needed.

The net gain achieved in our work exceeds 10 dB over the amplification bandwidth (essentially, this is how we defined the bandwidth), and 25 dB is the net gain achieved at the maximum of the parametric gain lobes.

Action taken: The discussion section of the manuscript has been updated such that the gain values are stated clearly to avoid confusion.

8. Although these results are impressive, the authors' anticipation of the use of these amplifiers as "ideal candidates for driving next-generation fiber systems based on doubly anti-resonant hollow-core fibers" seems premature. Given the strong non-linearity of the waveguide, strong FWM interactions between the signals are expected and need to be verified before such claims can be made.

We are thankful that the referee finds our results impressive. Based on published work on fiber-based parametric amplifiers, we expect FWM interactions between different signals not to be pronounced given that the efficiency of nonlinear processes scales with power, and the power of interacting signals is typically far lower than the power of the pump wave. Nevertheless, we agree that we should not have made this claim without proper measurements and verification. That said, we do believe that with further improvements, and maturing of the GaP photonic integrated circuits, such compact, ultra-broadband, and high gain on-

chip amplifiers could serve in applications in integrated photonic systems, as well as in cases where one has low light levels to amplify.

Action taken: The claim of potential usage of GaP parametric amplifiers for driving next-generation fiber systems based on doubly antiresonant hollow-core fibers has been removed.

Submission of a revised version of our manuscript “An ultra-broadband photonic-chip-based ~~traveling-wave~~ parametric amplifier”

We are grateful to the referees for their time and effort in reviewing the revised version of our manuscript and are delighted by their positive evaluation of our work.

Here we present a point-by-point response to the review. The original text of the report is printed in black, our replies are **highlighted in blue**, and the **action taken in red**.

Summary of reply to reviewers:

Referee #1 (Remarks to the Author):

The authors have answered all my questions very well, and made changes to the paper accordingly. Technically, I don't agree 100% with the authors but I do believe the quality of the paper meets the publication level. The paper should be published in my opinion.

We are glad we could address the questions and provide a comprehensive response, and we appreciate the positive recommendation made by the referee.

Referee #2 (Remarks to the Author):

Dear editor, I am satisfied with most of the authors' responses and recommend the publication of the paper. However, I remain unconvinced regarding the use of the term "traveling wave amplifier." The authors argue that "it is more important to highlight the physics of the process rather than the platform," but this is precisely my point. These amplifiers rely on phase-matched four-wave mixing processes in $\chi(3)$ media, regardless of the platform. Therefore, I maintain that this is more accurately described as a "parametric amplifier," which would be more familiar and clear to the optics community focused on this field. Referring to it simply as a "parametric amplifier" would not detract from the novelty and would enhance clarity.

We thank the referee for considering our revised paper and greatly value the recommendation for its publication. We now agree that changing the description to simply "parametric amplifier" improves clarity, as the term "traveling wave", which was used to differentiate these devices from resonant OPAs, is redundant, particularly in the title, and creates more confusion than it helps to avoid. Indeed, even in internal discussions, we typically refer to these devices simply as "parametric amplifiers" or OPAs. Accordingly, we have made the necessary changes throughout the manuscript, including changes in the title, and we mention once in the introduction that the term "traveling wave" is also used sometimes in literature, particularly in the microwave domain.

Action taken: the title has been changed to “An ultra-broadband photonic-chip-based parametric amplifier”; corresponding changes have been implemented in the text.